# Mesenchymal stem cells offer a drug-tolerant and immune-privileged niche to *Mycobacterium tuberculosis*

Neharika Jain[1], Haroon Kalam[1], Lakshyaveer Singh[1], Vartika Sharma[1], Saurabh Kedia[2], Prasenjit Das [3], Vineet Ahuja[2] & Dhiraj Kumar [1✉]

Anti-tuberculosis (TB) drugs, while being highly potent in vitro, require prolonged treatment to control *Mycobacterium tuberculosis* (*Mtb*) infections in vivo. We report here that mesenchymal stem cells (MSCs) shelter *Mtb* to help tolerate anti-TB drugs. MSCs readily take up *Mtb* and allow unabated mycobacterial growth despite having a functional innate pathway of phagosome maturation. Unlike macrophage-resident ones, MSC-resident *Mtb* tolerates anti-TB drugs remarkably well, a phenomenon requiring proteins ABCC1, ABCG2 and vacuolar-type H$^+$ATPases. Additionally, the classic pro-inflammatory cytokines IFNγ and TNFα aid mycobacterial growth within MSCs. Mechanistically, evading drugs and inflammatory cytokines by MSC-resident *Mtb* is dependent on elevated PGE2 signaling, which we verify in vivo analyzing sorted CD45$^-$Sca1$^+$CD73$^+$-MSCs from lungs of infected mice. Moreover, MSCs are observed in and around human tuberculosis granulomas, harboring *Mtb* bacilli. We therefore propose, targeting the unique immune-privileged niche, provided by MSCs to *Mtb*, can have a major impact on tuberculosis prevention and cure.

[1] Cellular Immunology Group, International Centre for Genetic Engineering and Biotechnology, Aruna Asaf Ali Marg, New Delhi 110067, India. [2] Department of Gastroenterology, All India Institute of Medical Sciences, New Delhi 110012, India. [3] Department of Pathology, All India Institute of Medical Sciences, New Delhi 110012, India. ✉email: dhiraj@icgeb.res.in

**M**ycobacterium tuberculosis (Mtb) continues to infect, cause illness (tuberculosis) and kill a large number of individuals globally. Among numerous factors that thwart any tuberculosis control program, lack of an effective vaccine and long duration of treatment are the two most critical ones[1,2]. Long treatment duration is majorly attributed behind noncompliance and emergence of drug-resistant tuberculosis including multi- and extensively drug-resistant (MDR and XDRs, respectively) ones[1]. While standard-of-care anti-TB drugs are very efficient in killing Mtb in liquid culture and during ex vivo infection studies in macrophages, their efficacy is dramatically compromised during in vivo infection studies and in the clinical practices, requiring prolonged treatment duration[3]. It is believed that Mtb undergoes metabolic adaptations within host granulomas, which render these bacteria less vulnerable to the standard drugs[4,5]. Driving factors, which cause such adaptations include nitric oxide (NO), redox stress (ROS), low oxygen (hypoxia), low nutrients, or altered carbon source[4,6–11].

Curiously, whatever we know about the intracellular lifestyle of mycobacteria in the hosts is mostly through studies on macrophages[12,13]. Are there additional niches of mycobacteria in vivo, which could facilitate the perceived metabolic adaptations? While there is no clear answer to the above assumption, there are certainly different other cell types which get infected inside the host including lung epithelial cells, macrophages, neutrophils, dendritic cells, adipocytes, and mesenchymal stem cells (MSCs)[14–18]. MSCs are peculiar among these cells since they were first reported to dampen the host immunity against tuberculosis around the granulomas[19]. Subsequently, it was observed that these cells are the site of persistent or latent bacterial infection[20]. Interestingly, latent bacteria are perceived to be more tolerant of anti-TB drugs[21–23]. Moreover, MSCs are classically known for their immune-modulatory functions[24–26]. Whether MSCs provide a privileged niche to mycobacteria allowing them to withstand the drug and evade host immunity remains unclear. Potential benefits mycobacteria enjoy within these cells continue to remain obscure due to lack of systematic studies on the intracellular lifestyle of Mtb within MSCs. MSCs can be readily isolated from bone marrow (animals) and adipose tissues (humans) thereby serving as an excellent ex vivo model to study mycobacterial lifestyle in these cells. In this study, using human adipose tissue-derived mesenchymal stem cells (ADSCs), we show that Mtb not only escapes the effect of anti-TB drugs while residing within ADSCs but also effectively evades host immune mediators. We further establish the mechanism behind these unusual properties of ADSCs and show their relevance during in vivo infection in mice as well as studies on the human granulomas.

## Results

### Adipose-derived mesenchymal stem cells (ADSCs) support mycobacterial growth.

Human primary adipose-derived mesenchymal stem cells obtained commercially were first characterized for the expression of cell-surface markers like CD73, CD44, CD90, CD105, CD271, and the negative marker CD11b (Supplementary Fig. 1a, b). Subsequently their ability to differentiate into three different lineages i.e., adipocytes, chondrocytes, and osteocytes were also characterized (Supplementary Fig. 1c–e). Next, we infected ADSCs with GFP-H37Rv (MOI 1:10) with ~80 percent efficiency (see Methods, Fig. 1a). Mean fluorescence intensity (MFI) measurements at 0, 3, 6, 9, and 12 days post infection showed that Mtb within ADSCs multiplied well (Fig. 1b), which we also confirmed by colony-forming unit (CFU) counts upon plating the bacteria released by lysing the infected ADSCs (Fig. 1c). A time-course growth analysis using CFU counts showed a massive increase in Mtb CFU at 9 and 12 days post infection (Fig. 1c). Consistent with previous reports from several groups including ours[27–29], H37Rv survived well in human primary macrophages and THP-1 derived macrophages (Fig. 1d, e, respectively); however, its multiplication within macrophages was markedly subdued when compared with that observed within ADSCs (Fig. 1). The vaccine strain BCG showed a marked decline in survival within ADSCs by 3 days post infection (Fig. 1f), which was also true in THP-1-derived macrophages (Fig. 1g). Infection with H37Rv did not result in spontaneous differentiation of ADSCs to any of the three lineages mentioned above (Supplementary Fig. 1c–e). A micro-array analysis of ADSCs infected with H37Rv for 6 days showed significant regulation of genes belonging to usual functional classes like immune regulation, inflammation, response to stress, transport pathways, and cholesterol metabolism etc. (Supplementary Fig. 2a).

### ADSC-resident Mycobacterium tuberculosis shows drug-tolerant phenotype.

Since MSCs were reported to serve as a site for bacterial persistence[20] we were keen to understand how Mtb residing within these cells responds to anti-TB drugs. We treated Mtb-infected ADSCs with different doses of isoniazid (INH; 0.5, 1, and 5 µg/ml) or rifampicin (RIF; 0.1, 0.5, and 5 µg/ml) for 24 h before harvesting the cells and CFU plating on 3rd, 6th, 9th, and 12th days post infection. Even at doses as high as 5 µg/ml for INH, ~5% of Mtb survived after drug treatment (Fig. 1h). At 0.5 µg/ml as well as at 1 µg/ml of INH nearly 10–15% of Mtb did not get killed (Fig. 1h, Supplementary Fig. 2b). In case of RIF, almost 50% of bacteria survived at 0.1 µg/ml dose, nearly 20–25% survived at 0.5 µg/ml, which did not go down below 15% even at doses as high as 5 µg/ml (Fig. 1h, Supplementary Fig. 2b). The percentage of bacteria, which survived the drugs, was considered as percent drug-tolerant bacteria. Interestingly, drug-tolerant phenotype (INH or RIF) of ADSC-resident Mtb was mostly independent of time spent within ADSCs as drug-tolerant H37Rv were observed as early as 3 days post infection and maintained at 6, 9, and 12 days post infection (Fig. 1i, j). At similar doses and for similar duration of treatment (i.e., 24 h) within human macrophages, there were hardly any surviving bacteria in case of INH (1 µg/ml, ~2–4%) while there were nearly 10–15% tolerant bacteria in case of RIF (0.5 µg/ml, Fig. 1i, j). These results suggest that ADSCs provide an environment, which allows Mtb to tolerate anti-TB drugs.

### Host ABC transporters ABCC1 and ABCG2 play a key role in bacterial drug tolerance.

MSCs are known to express a high level of ABC family transporters or efflux pumps, which are often attributed to drug tolerance in case of cancer[30,31]. In our microarray data, we did observe a marginal but consistent change in the expression of ABC transporters like ABCC1, also known as MRP1[32,33] (Supplementary Data 1). Among many ABC family transporters tested by quantitative RT-PCR, ADSCs showed an increase in the expression of ABCC1 and ABCG2 upon Mtb infection in MOI dependent manner (Supplementary Fig. 3a). Both intracellular as well as surface expression of ABCC1 and ABCG2 were higher in H37Rv-infected ADSCs with respect to the control cells (Fig. 1k). To test whether ABCC1 and ABCG2 were involved in imparting drug tolerance, we used known pharmacological inhibitors against them. Treatment with novobiocin (an ABCG2 inhibitor) or with MK571 (ABCC1 inhibitor) led to a decline in the drug-tolerant Mtb population (Supplementary Fig. 3b, c). Novobiocin, however, is also a well-known DNA gyrase inhibitor[34,35] and it could kill Mtb even in vitro in liquid cultures (Supplementary Fig. 3d). Unlike novobiocin, MK571 treatment did not have any effect on Mtb growth in vitro

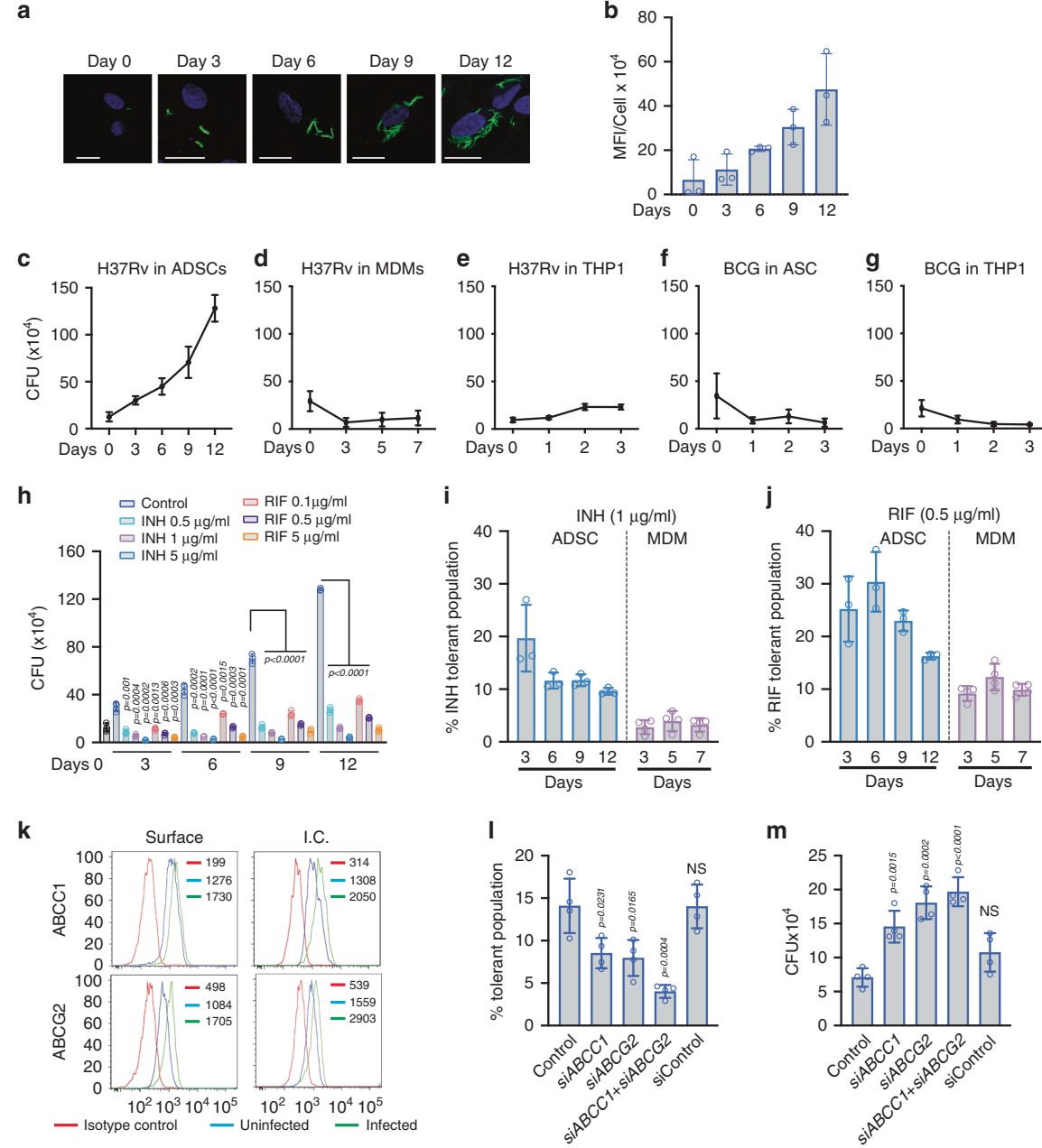

**Fig. 1 ADSCs support better *H37Rv* survival and high drug tolerance. a** Representative confocal images of *GFP-H37Rv*-infected ADSCs at 0 h, 3rd, 6th, 9th, and 12th days post infection (dpi), Scale bars, 20 µm. **b** MFI of *GFP-H37Rv*/cell across above mentioned time points is represented in bar graph as mean ± SD, n = 3 independent experiments, each representing five fields consisting of 25–100 cells. **c** Growth kinetics of *H37Rv* within ADSCs across 12 dpi. **d** Growth kinetics of *H37Rv* within human MDMs across 7 days post infection and (**e**) in THP-1 macrophages across 3 days. **f** Growth kinetics of *BCG* within ADSCs and THP-1 macrophages (**g**) across 0, 1, 2, and 3 dpi. **h** *H37Rv*-infected ADSCs were treated with doses of isoniazid (INH), and rifampicin (RIF) for 24 h prior to 3rd, 6th, 9th, and 12th dpi, and were plated for CFU enumeration. n = 3 donors, ****p < 0.0001. **i** Percent drug-tolerant bacterial population to INH (1 µg/ml) within ADSCs (blue) on 3rd, 6th, 9th, 12th days and within MDMs (purple) on 3rd, 5th, and 7th dpi, respectively (see Methods). n = 3 donors (ADSC), 4 donors (MDM). **j** Percent drug-tolerant bacterial population to RIF (0.5 µg/ml) within ADSCs (blue) on 3rd, 6th, 9th, and 12th days and within MDMs (purple) on 3rd, 5th, and 7th dpi, respectively. n = 3 donors (ADSC), 4 donors (MDM). **k** Line histogram of surface and intracellular (I.C.) staining of ABCC1/MRP-1 and ABCG2/BCRP in uninfected and in *H37Rv*-infected ADSCs, 6 dpi. Numbers represent MFI of individual colored histogram. **l** Percent INH (1 µg/ml) tolerant bacterial population in ADSCs after knocking down *ABCC1* (200 nM siRNA) or *ABCG2* (200 nM siRNA) alone or in combination, in *H37Rv*-infected ADSCs, n = 4 independent experiments. **m** *H37Rv* CFU from 6th day infected ADSCs after siRNA mediated knockdown of *ABCC1* and *ABCG2* for 48 h prior to the time point. n = 4 independent experiments. All data were analyzed by two-tailed unpaired Student's *t* test and represented as mean ± SD. Source data are included in the source data file.

(Supplementary Fig. 3e). Since pharmacological inhibitors may still have off-target effects, we knocked down these transporters using specific siRNAs. Knocking down either *ABCC1* or *ABCG2* led to a substantial decline in the drug-tolerant bacterial population within ADSCs (Fig. 1l and Supplementary Fig. 3f). When *ABCC1* and *ABCG2* both were knocked down simultaneously, the tolerant bacterial population declined further, reaching nearly 2–4 % (Fig. 1l). There was no such effect on drug-

tolerant population when scrambled siRNA was used (Fig. 1l). While the results with tolerant population did indicate the role of ABCC1 and ABCG2 in *Mtb* drug tolerance within ADSCs, in parallel experimental groups where no drug was used, knocking down *ABCC1* and *ABCG2* led to a considerable increase in bacterial CFU (Fig. 1m). The increase in bacterial CFU was more when both *ABCC1* and *ABCG2* were simultaneously knocked down, whereas there was no effect when scrambled siRNA was used as a control (Fig. 1m). Similar results were also obtained when ABCC1 or ABCG2 were inhibited by corresponding pharmacological inhibitors (Supplementary Fig. 3g, h).

**Role of lysosomal function in mycobacterial drug tolerance in ADSCs.** While ABCC1 and ABCG2 seemed important for drug tolerance, their role in bacterial killing as noted in Fig. 1m was surprising and indicated the presence of an active bactericidal mechanism in ADSCs. Interestingly inhibition of vacuolar-type H$^+$ ATPases by bafilomycin A1 (BafA1) also dramatically reduced the drug-tolerant *Mtb* within ADSCs (Fig. 2a). Another lysosomal acidification inhibitor chloroquine (CQ) had a similar effect (Fig. 2a). However, conditions, which led to increased lysosomal maturation like rapamycin treatment showed drug-tolerant population at par with the control ADSCs (Fig. 2a). Since rapamycin is a well-known inducer of autophagy[36], we also verified it using autophagy inhibitor 3-methyladenine (3MA), which expectedly led to a decline in the drug-tolerant population (Fig. 2a). Thus lysosomal function was probably essential to achieve drug-tolerant phenotype within ADSCs. Interestingly, in the absence of INH, conditions, which resulted in reducing the drug-tolerant population, helped bacterial survival. Thus, BafA1, 3MA, and CQ treatment resulted in increased bacterial CFU, whereas rapamycin treatment led to a decline in the CFU suggesting a role of lysosomal killing mechanism in MSCs (Fig. 2b). The similarity in the phenotype observed by lysosomal acidification inhibitors and ABCC1/ABCG2 inhibition raises the possibility that the role of ABC proteins observed here could have more to do with the lysosomal function rather than the efflux activity at the cell surface. Before further exploring into the mechanism of drug tolerance within ADSCs, we wanted to compare this phenomenon with the reported instances of drug tolerance in macrophages[37].

**Effect of inflammatory cytokines IFNγ and TNFα on drug tolerance within ADSCs.** In macrophages, activation with inflammatory cytokines is known to induce drug-tolerant phenotype of *Mtb*[37]. We reconfirmed, in THP-1 macrophages, treatment with IFNγ or TNFα led to a substantial increase in the drug-tolerant population from ~3–4% in control to 30–40% in the activated cells (Fig. 2c). At similar doses of these cytokines, in the absence of drug, nearly 50% of the bacteria got killed, in agreement with the antibacterial state these cytokines impart to the activated macrophages (Fig. 2d)[37]. In ADSCs, IFNγ treatment at 5, 12.5, and 25 ng/ml led to an increase in the INH-tolerant population (Fig. 2e). In the case of TNFα treatment, INH-tolerant *Mtb* population was significantly higher at 20 ng/ml (Fig. 2e). More startling observation, however, was the case where *Mtb*-infected ADSCs were treated with these cytokines in the absence of drugs. There was nearly dose-dependent increase in the bacterial CFU upon treatment of *Mtb*-infected ADSCs with IFNγ or TNFα (Fig. 2f). The pro-bacterial effect of IFNγ and TNFα on *Mtb*-infected ADSCs was specific to the stimulus and corresponding downstream signaling since upon neutralization with purified IFNγR1 or with anti-TNFα antibody, we could revert the pro-bacterial effect of IFNγ or TNFα stimulation on bacterial CFU (Fig. 2g). Expectedly, with a similar neutralization

experiment in THP-1-derived macrophages, there was a rescue of *Mtb* from cytokine-mediated killing (Fig. 2h). Thus *Mtb*-infected macrophages and MSCs respond in a contrasting manner to IFNγ and TNFα stimulus.

**Analysis of intracellular niches of *Mtb* shows classic phagosome maturation dynamics in ADSCs.** Results from conditions like *ABCC1* or *ABCG2* knockdown, BafA1 treatment or IFNγ or TNFα treatments, all of which led to an increase in the bacterial survival suggest that ADSCs, despite supporting robust growth of the bacteria, keep actively killing the bacilli. To check whether phagosome maturation pathways as observed in macrophages are operational in a similar fashion during *Mtb* infection in ADSCs, we assayed for *Mtb* colocalization with early phagosomes (RAB5), late phagosomes (RAB7), lysosomes (LAMP1), and acidified lysosomes (LAMP1 and LysoTracker). At any given time post infection, a large number of bacteria (~40–50%) stayed within RAB5 positive early phagosomes inside the ADSCs (Supplementary Fig. 4a). While only 2–5% of bacteria were ever present in the RAB7 positive late phagosomes (Supplementary Fig. 3b). A similar distribution of *Mtb* is also reported within macrophages by others and also confirmed by us[38] (Supplementary Fig. 3c). This reflects the phagosome maturation arrest inflicted by *Mtb* in the infected macrophages[38,39]. However, unlike macrophages where LAMP1-*Mtb* or Lysotracker-*Mtb* colocalization rarely crosses ~15%[27], there are more bacteria (~30–40%) present in LAMP1 or Lysotracker-LAMP1 double-positive compartments in ADSCs (Fig. 2i, j, Supplementary Fig. 4d). The matured lysosomes, i.e., LAMP1 positive acidified compartments indeed reflect the killing mechanism in ADSCs since ~80% of intracellular BCG, the strain that gets killed within ADSCs, are present in Lysotracker-LAMP1 double-positive compartments (Fig. 2k). Interestingly, treatment with IFNγ, TNFα, or MK571, in general, led to a decline in bacterial localization to LysoTracker or LAMP1 + LysoTracker compartments; however LAMP1 compartment alone showed only marginal decline (Fig. 2j, Supplementary Fig. 4d). Exclusion of Cathepsin D (CatD), the lysosomal protease, from LAMP1-Lysotracker-positive compartment strongly correlated with increased bacterial survival upon IFNγ- or MK571-treated cells (Fig. 2l, Supplementary Fig. 4d). Interestingly, ABCC1 and ABCG2 were also found to co-localize with *Mtb* suggesting their recruitment to the phagosomes (Supplementary Fig. 4e). At least in the case of TNFα treatment, exclusion of ABCC1 from the LAMP1-LysoTracker compartment was highly significant, whereas IFNγ or MK571 treatment showed only marginal decline (Fig. 2m) leaving an impression that ABCC1 is probably directly involved in bacterial killing. However, their direct role in *Mtb* killing is supported by only one set of evidences—increased bacterial survival upon their knockdown or inhibition. The strong correlation between bacterial killing and their colocalization to LAMP1-Lysotracker-CatD compartment nonetheless suggest an active role of canonical phagosome maturation pathways in the bacterial killing within ADSCs.

Interestingly, in contrast to what is known in macrophages[27], *Mtb* very rarely co-localized to the autophagosomes and xenophagy flux was completely absent in ADSCs (Supplementary Fig. 5a), suggesting little role if any, of autophagy in controlling *Mtb* within ADSCs. Curiously, ADSCs showed very high basal autophagy flux (Supplementary Fig. 5b), which is critical for maintaining the stem cell like property of these cells[40], highlighting the segregation of homeostatic, and antibacterial arms of autophagy as reported by us earlier[41]. Moreover, treatment with IFNγ led to increased autophagy flux in ADSCs, unlike what was reported previously for macrophages (Supplementary Fig. 5c)[42]. Unlike macrophages, IFNγ treatment had no effect on cellular ROS generation in ADSCs (Supplementary Fig. 5d). Combined with the colocalization results

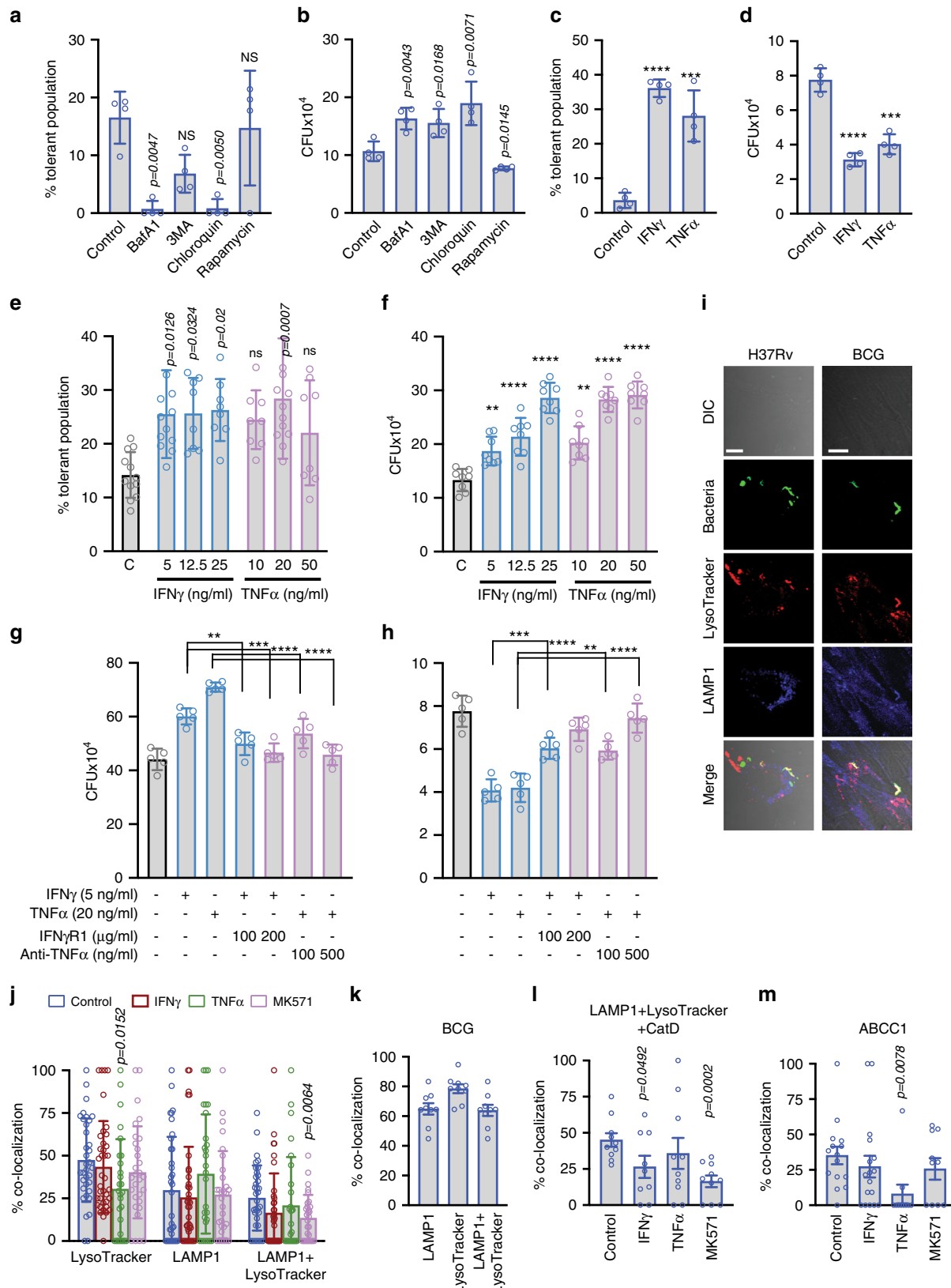

discussed above, these results largely explain why IFNγ failed to induce killing of *Mtb* within ADSCs.

**The lipid mediator PGE2 helps MSCs exhibit pro-bacterial attributes**. Results so far establish that MSCs are uncharacteristically pro-bacterial, at least during mycobacterial infections, helping them evade anti-TB drugs as well as classic host immune mediators like IFNγ and TNFα. However, there still was no clue on how MSCs execute these behaviors. To understand the mechanistic basis of the observed results, we went back to our microarray

**Fig. 2 Lysosomal killing of bacteria in ADSCs and effect of inflammatory cytokines. a**, **b** H37Rv CFU from infected ADSCs after bafilomycin A1 (BafA1, 100 nM), 3-methyladenine (3MA, 5 mM), chloroquine (100 μM), or rapamycin (100 nM) treatment for the final 6 h in INH treated (24 h) (**a**) or untreated (**b**) groups, n = 4 (**c**, **d**) 3rd day CFU from H37Rv-infected dTHP-1 treated with IFNγ (5 ng/ml) or TNFα (20 ng/ml) for 24 h in the presence (**c**) or absence (**d**) of INH, n = 4. Dose-dependent effect of IFNγ (5, 12.5, 25 ng/ml) and TNFα (10, 20, 50 ng/ml) treatment (24 h) on H37Rv survival in the infected ADSCs in the presence (**e**) or absence (**f**) of INH, n = 4 independent experiments with 2–3 biologically independent samples. CFU assay of infected ADSCs (**g**) and infected THP-1 macrophages (**h**) upon IFNγ and TNFα treatment along with doses of IFNγR1 and anti-TNFα purified proteins, respectively, n = 5. **i** Representative confocal images of PKH67-labeled H37Rv- and BCG-infected ADSCs (3rd dpi), stained with LysoTracker Red and LAMP1. Scale bars, 10 μm. Images are representative of >50 fields across three independent experiments. Percent colocalization of H37Rv with LysoTracker Red, LAMP1, LysoTracker + LAMP1 (**j**, n = 5, 5–8 independent fields each) and CatD + LAMP1+lysotracker triple positive (**l**, n = 3, 3–4 independent fields each) in ADSCs which were either untreated control or treated with IFNγ (5 ng/ml), TNFα (20 ng/ml) or MK571 (50 μM) for 24 h prior to samples processing at 3rd dpi, **k** Percent colocalization of BCG with LysoTracker Red, LAMP1, and LAMP1 + LysoTracker compartments at 3rd dpi in ADSCs, n = 3, three independent fields each. **m** Percent colocalization of H37Rv with ABCC1 in ADSCs control or IFNγ, TNFα, MK571 treated 3rd day samples, n = 3, 3–5 independent fields each. **a**–**h**: mean ± SD, **j**–**m**: mean ± SEM. Data were analyzed using two-tailed unpaired Student's t test (**b**, **d**, **f**, **j**–**m**) and one-way ANOVA (**a**, **c**, **e**, **g**, **h**). *p < 0.05, **p < 0.005, ***p < 0.0005, ****p < 0.0001, NS 'not significant' Source data are included in Source data File.

data to identify genes showing significant regulation upon Mtb infection in ADSCs. The anti-inflammatory as well as immune-modulatory functions of MSCs are well known; however, in all such known cases, MSCs execute its role by modulating functions of other cells, including T cells and macrophages[43–45]. Some of the key mediators that help MSCs execute these functions are PGE2, IDO1, IL6, CCL2, VEGFC, LIF etc.[24,45–49]. In our microarray data, genes from the PGE pathway like PTGS2, PTGES, and PTGR2 showed nearly eight-, four-, and fourfolds (log$_2$) increase in expression, respectively (Supplementary Fig. 6a). Similarly, IDO1 showed sixfolds increase, whereas LIF, IL6, CCL2, and VEGF each showed more than threefold increase in expression post infection (Supplementary Fig. 6a). We first tested PGE2 levels in the culture supernatants of ADSCs that were infected with H37Rv. Consistent with the microarray data PGE2 ELISA confirmed increased synthesis and secretion of PGE2 from Mtb-infected ADSCs (Fig. 3a). This was true for ADSCs obtained from multiple independent donors (Fig. 3a). Interestingly, treatment with IFNγ or TNFα further increased PGE2 levels in the culture supernatants whereas MK571-treated cells showed almost similar level of PGE2 as infected control cells (Fig. 3a). We used celecoxib, a widely used PTGS2 (or COX2) inhibitor, which is also an FDA approved drug in the market, as a negative control. Mtb-infected ADSCs treated with celecoxib showed negligible PGE2 levels by ELISA (Fig. 3a). Next, we treated Mtb-infected ADSCs with celecoxib at 50, 150, and 250 μM concentrations under all the conditions tested so far in this study. Treatment with celecoxib reduced Mtb CFU in a dose-dependent manner across the conditions including infection alone or when treated with IFNγ, TNFα, or MK571 (Fig. 3b). This result was tested on ADSCs from three independent donors and each of them showed similar results (Fig. 3b). Similar results were also obtained with EP2 receptor (receptor for PGE2) antagonist PF04418948, suggesting the involvement of signaling through the PGE2 pathway in bacterial survival (Supplementary Fig. 6b). Celecoxib was also effective in killing Mtb within macrophages however unlike in ADSCs, it did not show any dose-dependent killing in macrophages at the tested doses (Supplementary Fig. 6c). Unlike ADSCs, there was no increase in PGE2 release by THP-1 macrophages upon infection or treatment with IFNγ or TNFα (Supplementary Fig. 6d). We also verified these results by knocking down PTGS2 (COX2) using specific siRNAs (Fig. 3c and Supplementary Fig. 6e). Confocal microscopy revealed that a majority of bacteria in celecoxib-treated cells or PF04418948-treated cells or COX2 siRNA-treated cells co-localized with LAMP1, Lysotracker as well as CatD (Fig. 3d and Supplementary Fig. 6f). Interestingly, COX2 inhibition by celecoxib also helped limit the drug-tolerant phenotype in ADSCs against INH, irrespective of treatment with IFNγ or TNFα (Fig. 3e). Moreover,

MK571, which itself decreases drug-tolerant Mtb in ADSCs when combined with celecoxib treatment further reduces the drug-tolerant population of Mtb within ADSCs (Fig. 3e). The effect of celecoxib on bacterial drug tolerance was PGE2 mediated and not due to a possible role of certain COX2 inhibitors directly on bacterial drug-resistance protein MDR1[50] since knocking down COX-2 also led to a remarkable decline in INH tolerant as well as rifampicin-tolerant Mtb population within ADSCs (Fig. 3f, g, respectively). Efficacy of COX2 knockdown by siRNA on bacterial drug tolerance also rule out the role of PGE2 inhibitors in directly regulating the efflux proteins as reported previously[51].

**MSCs serve as a niche for Mtb during in vivo infection allowing drug tolerance in PGE2 dependent manner.** While all the results so far were performed on human primary adipose tissue-derived mesenchymal stem cells, we next wanted to explore whether these cells get involved during in vivo infection in mice and humans as well as to know whether PGE2 signaling plays a similar role in vivo. We infected C57BL/6 mice with H37Rv through aerosol challenge and 4 weeks post infection, these animals were divided into four groups: control, celecoxib (50 mg/kg), INH (10 mg/kg), or INH + celecoxib (10 and 50 mg/kg, respectively); and treated for subsequent 4 and 8 weeks. We deliberately used lower INH doses (10 mg/kg instead of 25 mg/kg) in order to observe the additive effect of celecoxib during combination treatment. From the initial bacterial load of 100–150 per animal, it reached around $2 \times 10^6$ per animal by the end of 4 weeks, ~$3 \times 10^6$ by the end of 8 weeks and ~$5 \times 10^6$ by the end of 12 weeks in the lungs (Fig. 4a). While celecoxib treatment alone did not significantly reduce the bacterial load at 4 weeks or 8 weeks of treatment (Fig. 4a). INH treatment brought the bacterial CFU significantly down at 8 weeks post treatment (Fig. 4a). Animals, which received both celecoxib and INH showed a more significant reduction in bacterial CFU in the lungs at both 4 and 8 weeks post treatment (Fig. 4a). Similar results were also obtained in the spleen (Fig. 4b). The combination treatment was significantly more effective with respect to INH or celecoxib alone in controlling bacterial load in both lung and spleen at 8 weeks of treatment (Fig. 4b).

We next sorted lung tissues from the infected animals at each time points and treatment groups into CD45−Sca1+CD73+ (MSCs) and CD45+Ly6G−CD11B+ (macrophages) cells with a purity of more than 90% (Fig. 4c and Supplementary Fig. 7a). The sorted population was further characterized with additional cell-surface markers for macrophages and MSCs, respectively. Thus CD11b+ cells were confirmed LY6G−, CD11c$^{low}$, Ly6C+, and MHCII+ (Fig. 4d)[52]. Similarly, CD73+ cells were CD11b−, CD44+, CD90+, and CD105+ (Fig. 4d). The population of macrophages across all treatment groups declined from a

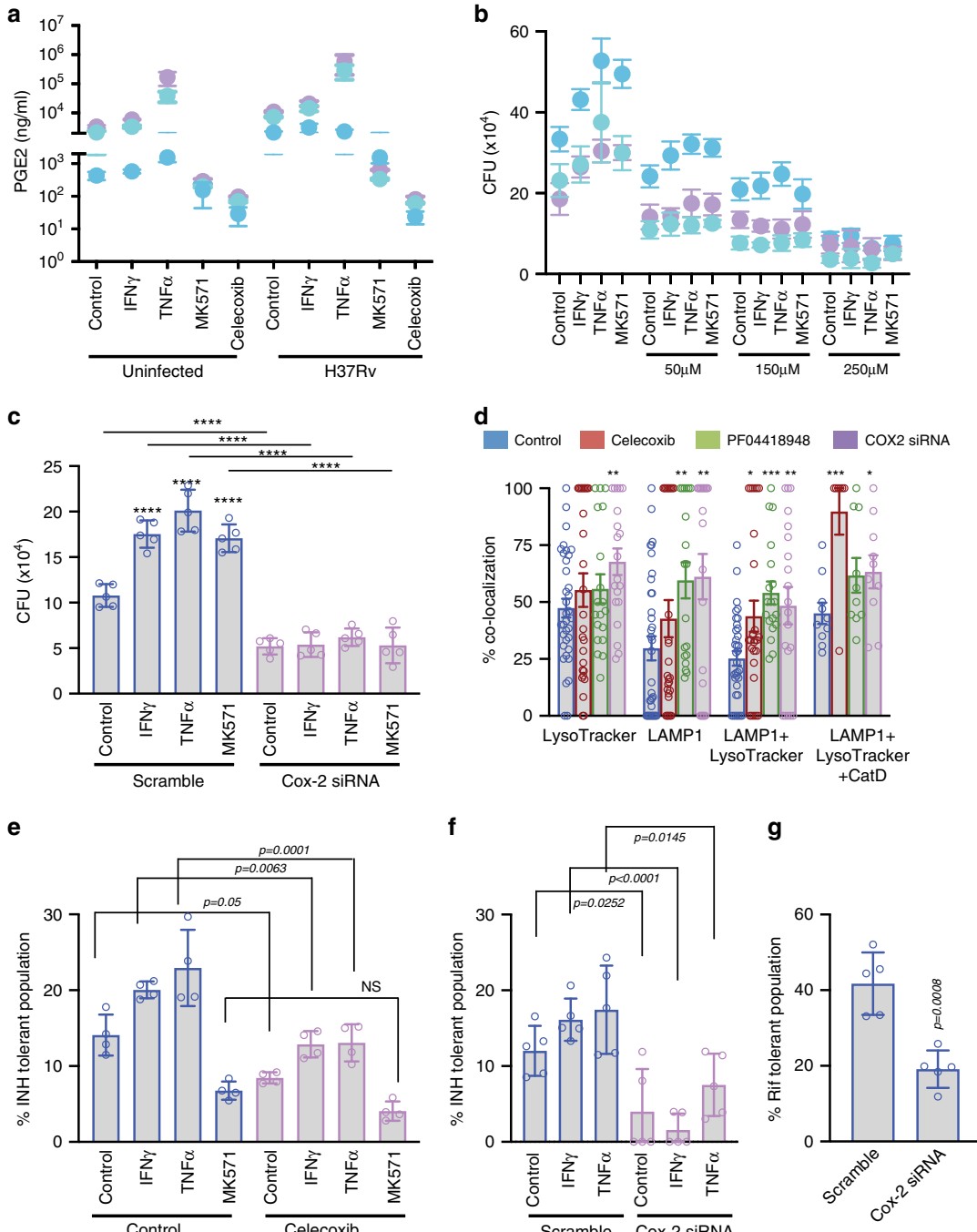

**Fig. 3 Lipid mediator PGE2 modulates lysosomal activity in *Mtb*-infected ADSCs. a** Uninfected or *H37Rv*-infected ADSCs were either untreated or treated with IFNγ, TNFα, MK571 and celecoxib (250 μM, negative control) for 24 h before collecting supernatant for ELISA on 6th dpi (6th day post seeding in the uninfected cells), Mean ± SD, *n* = 3 donors represented by three colors. **b** CFU enumeration of *H37Rv*-infected ADSCs treated with different doses of celecoxib (50 μm, 150 μM, 250 μM) in addition to IFNγ, TNFα, or MK571 for 24 h before plating on 6th dpi, Mean ± SD, *n* = 3 donors represented by three colors. **c** *H37Rv*-infected ADSCs treated with Cox-2 siRNA or scramble siRNA (100 nM) along with 24 h treatment with IFNγ, TNFα, or MK571 prior to CFU plating on 6th dpi, Mean ± SD, *n* = 5. **d** Percent Colocalization of *H37Rv* with LysoTracker Red, LAMP1, and Cathepsin D stained compartments in ADSCs left untreated or treated for 24 h with Celecoxib (250 μM), PF04418948 (500 nM), or Cox-2 siRNA (100 nM, 48 h) before fixing samples on 3rd dpi, Mean ± SEM, *n* = 3. **e**, **f** Percent INH-tolerant population in ADSCs treated with celecoxib (**e**, *n* = 4) or Cox-2/scramble siRNA (**f**, *n* = 5) along with IFNγ, TNFα or MK571 addition for 24 h prior to CFU plating on 6th dpi, Mean ± SD. **g** Percent RIF-tolerant bacterial population in infected ADSCs treated with scrambled or cox-2 siRNA (100 nM) for 48 h prior to CFU plating on 6th dpi, Mean ± SD, *n* = 5. Data were analyzed using two-tailed unpaired Student's *t* test (**a**, **c**, **d**, **f**, **g**) and one-way ANOVA (**b**, **e**). *$P < 0.05$, **$P < 0.005$, ***$P < 0.0005$, ****$P < 0.0001$, NS 'not significant' Source data are included in the source data file.

maximum at 4 weeks to 8 weeks and subsequently to 12 weeks post infection (Fig. 4e). The population of MSCs was also highest at 4 weeks post infection, which declined subsequently at 8 and 12 weeks post infection (Fig. 2f). In control uninfected animals,

macrophage and MSC population was relatively lower than the infected ones at either 4 or 12 weeks post infection (Supplementary Fig. 7b). Lysing and plating these sorted cells on 7H11 media showed the presence of *Mtb* in both macrophages and MSCs

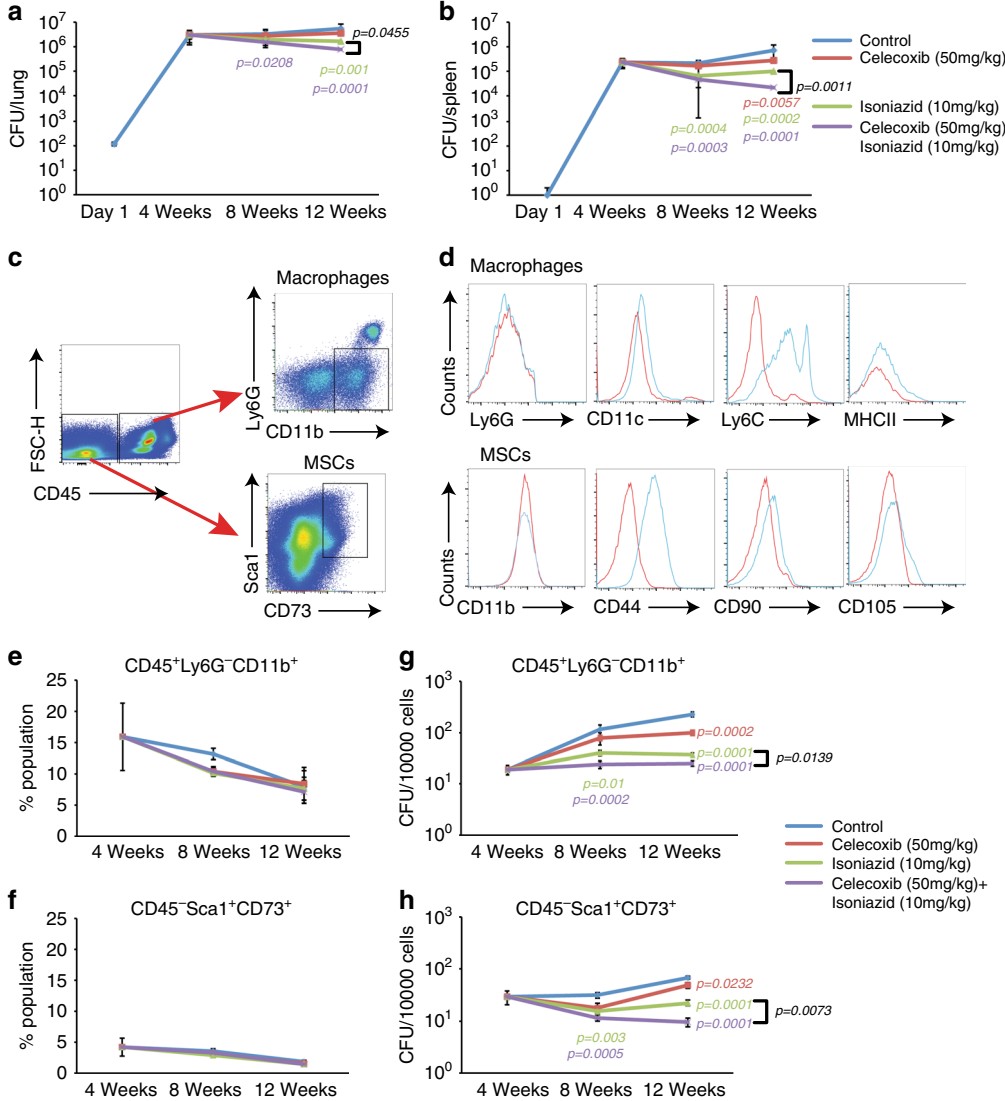

**Fig. 4 PGE2 facilitates *Mtb* survival within lung MSCs in vivo. a** Total bacterial CFU in the lungs of C57BL/6 mice infected with *H37Rv* via aerosol route (~$10^2$ bacilli/lung) administered with vehicle, celecoxib (50 mg/kg), INH (10 mg/kg), or combination of celecoxib with INH (50 mg/kg and 10 mg/kg, respectively). Treatments started 4 weeks post infection and were given every day for next 8 weeks. **b** Total bacterial CFU from the spleen of infected animals during the course of experiment mentioned above (**c**) Gating strategy for sorting of MSCs and monocyte/macrophages from mice lung. Live singlet population was gated for CD45 positive and negative population which were sub-gated based on Ly6G⁻CD11b⁺ for macrophages or myeloid cells and Sca1⁺CD73⁺ for MSCs, respectively. **d** Characterization of the gated macrophage and MSC population with additional cell specific surface marker. Upper panel is for CD11b⁺ macrophage stained for Ly6G, CD11c, Ly6C, and MHCII. Lower panel is for CD73⁺ MSCs stained with CD11b, CD44, CD90, and CD105. **e–f** Change in lung tissue landscape comprising macrophages (CD45⁺Ly6G⁻CD11b⁺) and MSCs across 12 weeks of infection and upon treatment with celecoxib, INH or celecoxib + INH. **g–h** *Mtb* survival within sorted macrophages (CD45⁺Ly6G⁻CD11b⁺) and MSCs (CD45⁻Sca1⁺CD73⁺) along the course of infection and treatment as discussed above. All above data are represented as mean ± SD, $n = 3$ independent experiments (total ten mice). Statistical analysis was done Mann−Whitney *U* test. Source data are included in the source data file.

(Fig. 4g, h). The number of bacilli in both cells progressively increased from week 4 to week 12, shown as the number of bacilli per 10,000 cells (Fig. 4g, h). The macrophage-resident *Mtb* were not affected by celecoxib at 4 weeks of treatment however showed a significant decline at 8 weeks post treatment (Fig. 4g). Similarly, the effect of INH was more pronounced in macrophages at 8 weeks post treatment (Fig. 4g). Animals which received both INH and celecoxib showed very significant decline in macrophage-resident *Mtb* at both 4 and 8 weeks post treatment, which was also marginally but significantly lower than INH alone group at 8 weeks post treatment (Fig. 4g). For MSC-resident *Mtb*, the effect of treatments was mostly similar to that observed in macrophages except for two interesting observations (Fig. 4g).

First, consistent with the ex vivo results, the effect of INH alone on MSC-resident *Mtb* was relatively less in magnitude when compared with that on macrophage-resident *Mtb* at 12 weeks post infection (Fig. 4h). Secondly, supplementing INH with celecoxib had a more dramatic effect on MSC-resident *Mtb* than macrophage-resident ones (Fig. 4g). Thus in vivo results from mouse mostly followed the observations obtained from ex vivo studies with human ADSCs.

**MSCs are present in human extra-pulmonary and pulmonary tuberculosis granulomas.** The results so far establish that MSCs serve as a niche for *Mtb* providing drug and immune-privileged niche, in PGE2 dependent manner both ex vivo and in vivo in

animals. We next analyzed the presence and spatial localization of CD73[+] cells with respect to *Mtb* in tissue sections from pulmonary and extra-pulmonary TB lesions from human subjects. CD73[+] cells were found in and around extra-pulmonary (gut) and pulmonary tuberculosis lesions (Fig. 5a, b, respectively). Intestinal biopsy samples were taken from granuloma-positive confirmed intestinal tuberculosis (ITB) patients[53]. In lung biopsies, in addition to CD73, we also stained for Ag85B of *Mtb* (Fig. 5b and Supplementary Fig. 7c). At lower magnification presence of CD73[+] cells in and around granulomas was distinctly visible (Fig. 5b). At higher magnification, we could observe presence of CD73-positive cells, which were also positive for Ag85B, suggesting presence of *Mtb* within these cells (Fig. 5c). We further confirmed the presence of *Mtb*-infected MSCs in lung granuloma sections by immunofluorescence using two different markers for MSCs-CD73 and CD105 (Fig. 5d, e and Supplementary Fig. 7d). All these experimental evidences conclusively demonstrate the presence of *Mtb*-infected MSCs in human tuberculosis granulomas.

## Discussion

Almost everything that we know about the intracellular lifestyle of *Mtb* largely emerged through studies on monocyte/macrophage models. The host responses and the mechanism of immune evasions are also studied keeping in mind macrophages as the primary cells where the bacteria reside[12,13]. The present study was undertaken to understand how MSCs could facilitate mycobacterial persistence in the host as reported by others[20]. This required us to explore the intracellular lifestyle of *Mtb* within MSCs, and not much is known about it. The immune-modulatory properties of MSCs are well known including during *Mtb* infection[19]. However, in a majority of cases, the immune-modulatory effects of MSCs are studied in trans i.e., on a different cell type, which is mediated by effectors released from MSCs[43–45]. Whether the innate ability of MSCs play a role in mycobacterial persistence and if these cells exhibit any cell-autonomous immune-modulatory properties, is not known. Interestingly, only virulent strain *H37Rv* could survive and divide well within ADSCs while BCG got killed, suggesting the presence of active innate defense mechanism in these cells. One critical aspect of mycobacterial persistence is tolerance to anti-TB drugs, which is driven by the host environment like macrophage residence, macrophage activation, low oxygen within granulomas, NO etc.[37,54]. Our finding that MSC-resident *Mtb* was tolerant to anti-TB drugs underscores the physiological advantage that these cells possess in order to harbor persistent infection as reported previously[20,55]. Since adult stem cells are known to have high efflux activity via ABC transporters, which helps in drug tolerance in cases like cancer[56,57] we questioned whether these efflux proteins could also help to throw out anti-TB drugs, thereby helping in drug tolerance. Our results indeed show increased expression and involvement of ABCC1 and ABCG2, in drug tolerance, both of which acted independently since their combined effects were greater than individual effects. However, inhibition of vacuolar-type H[+]ATPase by BafA1 led to a more dramatic decline in drug-tolerant population, suggesting phagolysosomal environment to be the key factor behind drug tolerance. Interestingly, phagosomal pH is also known to alter the redox physiology of *Mtb* leading to drug tolerance in macrophages, suggesting some convergence in the mechanism of bacterial drug-tolerance in MSCs and macrophages[58]. An increase in bacterial CFU upon ABCC1 or ABCG2 inhibition/knockdown in the absence of INH indicated the function of these proteins other than cellular efflux of drugs. Their recruitment to bacterial phagosomes indeed points to such a possibility especially since their recruitment to

the phagosomes largely correlated with bacterial killing. ABC proteins are known to have several moonlighting functions including nuclear translocation, redox balance and antigen presentation[59–62]. Results suggest, at least in MSCs, they are also involved in bacterial killing in the phagolysosomal system. Whether it is associated with lysosomal acidification or transport of bactericidal effectors remains to be uncovered. It is however, possible that ABC proteins are actively excluded from getting recruited to *Mtb* phagosomes while being present on other endolysosomal vesicles, where through their inward transport activities sequester certain antibacterial effectors including anti-TB drugs, away from mycobacterial phagosomes in isolated vesicles. This could potentially explain why knocking down or inhibition of ABCC1 or ABCG2 helps increased bacterial survival. At present we have no evidence to support whether these effectors could be H[+] ions, oxidized glutathione or glutathione metal adducts, ubiquitin-derived peptides, or any other antimicrobial peptides; each of which is capable of killing the bacteria and also known targets of ABC proteins-mediated transport across biological membranes[59,63].

Mycobacterial drug tolerance can also be induced in vitro or ex vivo in macrophages. In vitro, *Mtb* develops drug tolerance under stress conditions like hypoxia, NO, nutrient stress etc.[7,64]. There are also reports, which suggest mere macrophage residence for a few hours is sufficient to induce drug tolerance in *Mtb*[54]. Yet another study reported increased bacterial drug tolerance in activated macrophages[37]. Similarly, dependence of antibiotics on pH for bacterial killing is an emerging field of investigation[65]. More recently through a chemical screening approach, a compound was identified which help curtail bacterial tolerance to oxidative, acid and drug stress[66]. The common thread across these studies is that when *Mtb* witnesses stress whether in vitro or in vivo, it activates a set of genes which inadvertently also helps them tide-over the effect of drugs[11,37]. Thus pathways for stress tolerance and drug tolerance in *Mtb* seem to have some common players. In agreement with the activated macrophage studies, we noted further increase in drug tolerance of MSC-resident *Mtb* when activated by inflammatory cytokines like IFNγ and TNFα. Most surprisingly though, in the absence of drugs, IFNγ or TNFα treatment did not kill the bacteria, instead helped them grow better. To our knowledge, this is the first report, where under any circumstance a pro-bacterial role for these classic pro-inflammatory cytokines is reported. However, MSCs are known to show enhanced immune-modulatory properties when activated with inflammatory cytokines like IFNγ, TNFα, and even IL1β[67,68]. Some key antibacterial phenotypes activated in macrophages upon IFNγ stimulation include increased cellular ROS production, mitochondrial depolarization, autophagy inhibition etc.[42,69]. In MSCs, IFNγ treatment had no such effect and there was a significant increase in autophagy. Interestingly, unlike in macrophages, *Mtb* present in MSCs are not present in autophagosomes and therefore xenophagy flux is completely absent[27,41]. This partially explains the loss of antibacterial effects of IFNγ in ADSCs, although does not explain increased bacterial survival upon inflammatory stimuli. However similar to what we noted about ABCC1 or ABCG2 inhibition; pro-bacterial effects of IFNγ and TNFα had mostly to do with the lysosomal killing. Since each of these treatments was for the final 24 h before CFU plating was done, it cannot reflect increased bacterial replication rather show diminished bacterial killing. This observation however, brings an exceptionally worrisome insight on the problem of poor efficacy of every vaccine candidates tested so far. While there are several vaccine candidates at different stages of development against tuberculosis to replace or enhance BCG efficacy, the only commercially available vaccine, a closer look at each of the vaccine candidate shows that immunological parameters considered as

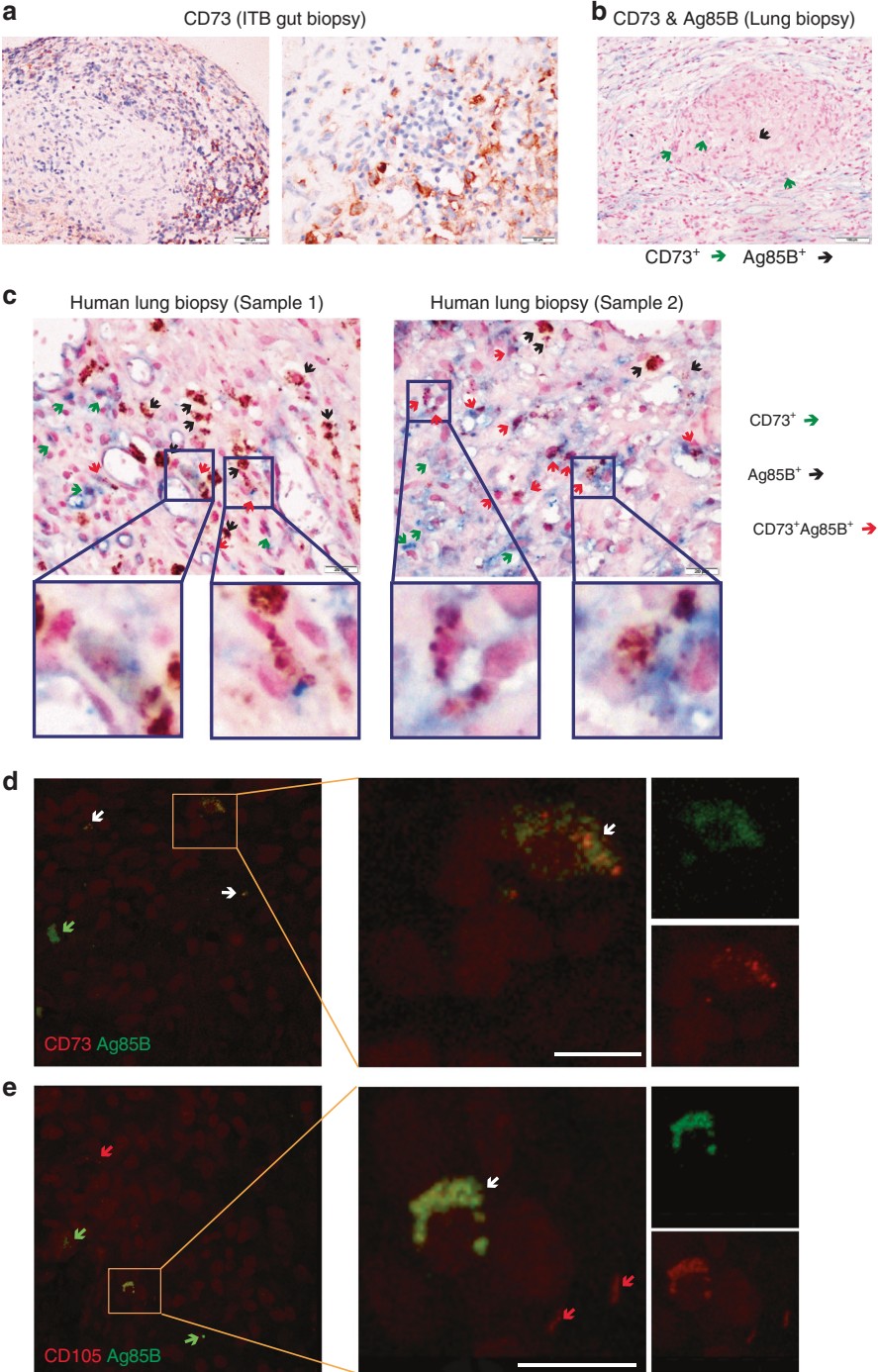

**Fig. 5 _Mtb_ co-localizes with MSC in human pulmonary and extra-pulmonary granulomas. a** CD73 staining of biopsies from granuloma-positive intestinal tuberculosis patient, showing polarization of CD73-positive cells around the submucosal macrogranulomas [×100 (left panel, scale bar = 100 μm), ×200 (Right panel, scale bar = 50 μm)]. **b** CD73 and Ag85B dual staining performed on lung biopsy tissue from patients with pulmonary tuberculosis showing polarization of CD73+ cells (green arrows) toward the granulomas. Ag85B-positive organisms (black arrows) are seen inside the granulomas (×100, scale bar = 100 μm). **c** Representative of two independent human lung biopsies, Ag85B-positive organisms (brown color) are seen inside the histiocytes (black arrows), CD73+ cells are stained with blue chromogen (green arrows) and the cells showing both positivity for CD73+ and Ag85B+ organisms have been represented by red arrows (×400, scale bar = 20 μm). Insets below show the corresponding magnified CD73+ cells showing positivity of Ab85B staining. **d** Immunofluorescence staining performed on formalin-fixed paraffin-embedded (FFPE) tissue of human lung biopsies from patients with known tuberculosis show green fluorescence for Ag85B+ (green arrows), red fluorescence for CD73+ cells (red arrows) and colocalization signals are marked with white arrows. The strong colocalization area is shown in the yellow inset and magnified in the panel at the right. In the further right panel, corresponding green and red channel fluorescence is shown. Scale bar is 10 μm. **e** FFPE tissue processed for dual IF staining show Ag85B+ only (green arrows), CD105+ MSCs (red arrow) and cells positive for both CD105 and Ag85B (white arrow). The strong colocalization area is shown in the yellow inset and magnified in the panel at the right. In the further right panel, corresponding green and red channel fluorescence is shown. Scale bar is 10 μm. Data shown in this figure are representative of seven independent experiments.

the correlates of protection are common across them[70]. Thus, whether it is MTBVAC or TB/FLU-04L, Ad5Ag85A, MVA85A, or others, they all rely on generating strong INFγ producing CD4+ and/or CD8+ T cells[71–73]. However given the unconventional pro-bacterial effects of IFNγ on MSCs, these vaccines can only generate an immune response that kills bacterial population in macrophages but not in MSCs thereby blunting the efficacy.

How Mtb enjoys such a privileged lifestyle within ADSCs became finally apparent through the microarray analysis revealing a massive increase in the synthesis and secretion of PGE2 by infected ADSCs. PGE2 is a multifunctional effector, with diverse roles in immune regulation[74]. PGE2-mediated immunomodulation of other cells by MSCs has also been extensively reported[48,49]. However here we report a unique autocrine immune-modulatory function of PGE2 in MSCs. Inhibiting PGE2 signaling was able to revert the pro-bacterial effects of IFNγ, TNFα, or MK571, suggesting PGE2 as the converging factor, which helps better bacterial survival within ADSCs. In contrast to the pro-bacterial role of PGE2 observed by us, several studies in the past report protective role of PGE2 against Mtb. Thus, loss of PTGES2 makes animals more susceptible to tuberculosis[75,76]. Similarly, EP2 receptor knockout mice also show increased bacterial burden in the lungs[77]. Interestingly, PGE2 treatment is more effective in controlling lung CFU only in hyper-susceptible animals lacking IL1R1[75], with absolutely no effect in WT animals. On the similar line, WT animals and ptgs2 animals (lacking the enzymatic activity) did not have any difference in bacterial survival[75]. On the other hand, during the late phase of mycobacterial infection and not during the early phase of infection, COX2 inhibition has protective effects in vivo[78]. PGE2 is also known to inhibit antibacterial effector functions of phagocytes including phagocytosis, NO production, lysosomal killing and antigen presentation[77]. Incidentally, aspirin is currently in the clinical trial as adjunct therapy against tuberculosis meningitis in adults[79]. Most COX2 inhibitors, specially nonsteroid anti-inflammatory drugs like aspirin, ibuprofen, rofecoxib, and celecoxib etc. are routinely used for controlling diverse inflammatory states[80]. Results from our experiments suggest smart inclusion of COX2 inhibitors in standard tuberculosis treatment/prevention regimens could enhance the efficacy of treatment. Presence of MSCs in human granulomas, in both pulmonary and extra-pulmonary tuberculosis cases, provides a valid basis to test such combinations for clinically favorable outcomes.

The two major hurdles in the tuberculosis control program are (a) lack of effective vaccine and (b) highly diminished efficacy of anti-TB drugs in vivo with respect to in vitro. This study shows MSCs contribute to both these crucial aspects of tuberculosis control. The remodeling of lung granulomas during tuberculosis has been explored previously[81]. However, we show that recruitment and infection of MSCs in the granulomas could be critical events during remodeling considering the lifestyle of Mtb in MSCs is radically different than those in macrophages. The study therefore also highlights the limitations of reliance on ex vivo data generated through macrophage infection experiments in the past. We believe targeting the immune-privileged environment of MSCs will help develop alternative strategies to enhance both treatment and vaccine efficacy.

## Methods

**Ethical clearance.** Studies on human samples were approved by IEC of AIIMS (Ref no. IEC-304/02-06-2017) and ICGEB (Ref no. ICGEB/IEC/2017/06-verII and ICGEB/IEC/2016/03). ITB biopsy samples were obtained after getting the informed consent from the patients. Human lung biopsy samples and control sections were obtained from Department of Pathology, AIIMS, New Delhi. These biopsy samples were taken for diagnostic purpose in these patients with written informed consent. Use of the archived leftover biopsy samples were approved by the institute's EC, as detailed above. Access to these materials is subject to the institutional guidelines.

Animal experiments were approved by Institutional Animal Ethics Committee, ICGEB (ICGEB/IAEC/280718/CI-14).

**Reagents, antibodies, and plasmids.** Phorbol 12-myristate 13-acetate (PMA), bafilomycin A1, rapamycin, 3MA, CQ, PKH, Mtb drugs (rifampicin, and pyrazinamide), chemical inhibitors (MK571, novobiocin, celecoxib, DMSO, BSA, MTT (1-(4,5-dimethylthiazol-2-yl)-3,5-diphenylformazan), and paraformaldehyde were obtained from Sigma Aldrich Co (St Louis, MO, USA). All IR conjugated secondary antibodies for immunoblotting were obtained from LI-COR Biosciences (Lincoln, NE, USA) while Alexa fluor conjugated secondary antibodies were procured from Invitrogen Molecular Probes, Carlsbad, CA, USA. PGE2 ELISA kit, isoniazid, propidium iodide (PI) were from Cayman Chemical, USA. Lysotracker red, JC-1, and cellrox green were from Life Technologies, USA. Human IFN-γ and human TNF-α were purchased from eBiosciences. siRNAs (ABCC1, ABCG2, COX2) were from GE DHARMACON. Safranin O, Oil Red O, and Alizarin Red S were purchased from SRL chemicals. All antibodies and their corresponding details are provided in the table below (Table 1).

**Cell culture.** ADSCs were purchased from life technology (cat no.—R7788115) and maintained in mesenPRO RS media (cat no. 12746012) supplemented with growth factors at 37 °C, 5% CO2, humidified incubator as per the manufacturer instructions and guidelines. For all in vitro experiments, ADSC were seeded at the required density, allowed to adhere to the surface for 24–36 h before proceeding with the experiment.

Human MDMs were isolated from human PBMCs, which in turn were separated from blood of healthy human donars using ficoll-layering and centrifugation. Briefly, heparinized blood was diluted in 1:1 ratio by volume with DPBS. Diluted blood was layered on Ficoll-paque (Himedia) and centrifuged at 2000 rpm for 45 min. Interface containing PBMC was isolated carefully and washed twice with DPBS. Cells were diluted in RPMI 1640 media containing 10% fetal bovine serum (FBS) to a concentration of $1 \times 10^6$ cells/ml. Cells were put in a six well tissue culture plate and incubated for 3 h in a humidified 5% CO2 chamber at 37 °C. Non adherent cells were removed followed by two washes with RPMI. Complete media containing 50 ng/ml recombinant human M-CSF (R&Dsystems, 216-MC/CF) was added and cells were allowed to differentiate for 7 days into macrophages in a humidified 5% CO2 chamber at 37 °C.

Human monocytic cell line THP-1 were obtained from American Type Culture Collection (ATCC) and cultured in RPMI 1640 medium along with 10% FBS at 37 °C, 5% CO2 humidified incubator. THP-1 derived macrophages (dTHP-1) were obtained by treating THP-1 cells with 20 ng/ml phorbol myristate acetate (PMA, sigma) for 24 h followed by washing and maintenance in complete media.

**MSC characterization.** MSC characterization into three lineages i.e., osteocytes, chondrocytes, and adipocytes were done according to the differentiation protocols provided by Life Technologies. In brief, MSCs were seeded in 24 well plates and after 6–8 h of adherence their media were replaced with adipocyte differentiation media (cat no. A1007001), chondrocyte differentiation media (cat no. A1007101), and osteocyte differentiation media (cat no. A1007201). Media were replaced every 3rd day till 21th days. After 21 days, cells were fixed and stained with Alizarin red S for chondrocytes, Oil Red O for adipocytes, Safranin O for osteocytes. In addition, differentiated ADSCs were also characterized using RT-PCR (chondrocytes and osteocytes) and FACS staining (adipocytes). Briefly, RNA was extracted from 21 days differentiated and undifferentiated ADSCs using mdi RNA Isolation kit. About 500 ng RNA was converted to cDNA using iscript cDNA synthesis kit, which was further used to set up real time PCR with RT primers for cell specific differentiated genes.

COL1A1 FP CCTGTCTGCTTCCTGTAAACTC
COL1A1 RP GTTCAGTTTGGGTTGCTTGTC
SOX9 FP GCAGCGAAATCAACGAGAAAC
SOX9 RP TCCAAACAGGCAGAGAGATTTAG
COL2A1 FP CACACTCAAGTCCCTCAACAA
COL2A1 RP AGTAGTCTCCACTCTTCCACTC
ACAN FP GACATTAGTGGGAGAGCTAGTG
ACAN RP GACACCAAAGAGTCCAGGTATT
RUNX2 FP CATCACTGTCCTTTGGGAGTAG
RUNX2 RP ATGTCAAAGGCTGTCTGTAGG
BMP-2 FP GAGAAGGAGGAGGCAAAGAAA
BMP-2 RP GGGACACGTCCATTGAAAGA
OPN FP CATATGATGGCCGAGGTGATAG
OPN RP AGGTGATGTCCTCGTCTGTA
OCN FP TCACACTCCTCGCCCTATT
OCN RP CCTCCTGCTTGGACACAAA

Adipocytes were confirmed using LipidTox staining as per the manufacturers protocol (Life technology, Cat no H34477) and analyzed using flow cytometry.

**Bacterial cultures and in vitro infection experiments.** Bacterial strains: H37Rv seed stock was received from Colorado State University. BCG (Danish strain) was obtained from University of Delhi, South Campus. For in vitro experiments, virulent laboratory strain H37Rv, BCG, and GFP-H37Rv bacterial cultures were

**Table 1 Details of antibodies used.**

| Serial No. | Antibody | Cat. no. | Company | IFA/FACS | Confocal | WB | IHC |
|---|---|---|---|---|---|---|---|
| | | | | colspan Application (dilution) | | | |
| 1 | MAP1LC3B | NB100-2220 | NovusBiological | | 1:200 | 1:1000 | |
| 2 | GAPDH | sc-48167 | Santa Cruz Biotechnology | | | 1:5000 | |
| 3 | COX2 | ab62331 | Abcam | | | 1:1000 | |
| 4 | Rab5 | sc-309 | Santa Cruz Biotechnology | | 1:200 | | |
| 5 | Rab7 | sc-6563 | Santa Cruz Biotechnology | | 1:200 | | |
| 6 | ABCC1 | sc-18835 | Santa Cruz Biotechnology | 1:100 | 1:100 | | |
| 7 | LAMP1 | sc-20011 | Santa Cruz Biotechnology | | 1:200 | | |
| 8 | ABCG2 | sc-58222 | Santa Cruz Biotechnology | 1:100 | 1:100 | | |
| 9 | Cathepsin D | ab19555 | Abcam | | 1:200 | | |
| 10 | CD73-FITC (clone AD2) | 561254 | BD Bioscience | 1:100 | | | |
| 11 | CD11b-PE | 557321 | BD Bioscience | 1:100 | | | |
| 12 | CD271 | ab8874 | Abcam | 1:100 | | | |
| 13 | CD44 | MS668P | Thermo Scientific | 1:100 | | | |
| 14 | CD90 | ab181469 | Abcam | 1:100 | | | |
| 15 | CD105 | ab114052 | Abcam | 1:100 | | | |
| 16 | CD45-APC (clone 30-F11) | 559864 | BD Bioscience | 1:100 | | | |
| 17 | CD73-BV450 (clone TY/23) | 561544 | BD Bioscience | 1:50 | | | |
| 18 | CD11b-APC-Cy7 (clone M1/70) | 557657 | BD Bioscience | 1:100 | | | |
| 19 | Ly6G−PE (clone 1A8) | 551461 | BD Bioscience | 1:100 | | | |
| 20 | Sca-1-PE-CF594 (clone D7) | 562730 | BD Bioscience | 1:100 | | | |
| 21 | CD90-BB515 (clone OX7) | 564607 | BD Bioscience | 1:100 | | | |
| 22 | CD44-BV605 (Clone IM7) | 563058 | BD Bioscience | 1:100 | | | |
| 23 | CD105-PE (Clone MJ7/18) | 562759 | BD Bioscience | 1:100 | | | |
| 24 | Ly6c-BV605 (Clone AL-21) | 563011 | BD Bioscience | 1:100 | | | |
| 25 | I-A/I-E-BB700 (clone 2G9) | 746086 | BD Bioscience | 1:100 | | | |
| 26 | CD11c-PE-Cy7 (clone HL3) | 558079 | BD Bioscience | 1:100 | | | |
| 27 | Rabbit anti-Ag85B | ab43019 | Abcam | 1:2000 | | | 1:2000 |
| 28 | Mouse anti-CD73 (1D7) | ab91086 | Abcam | 1:1600 | | | 1:500 |
| 29 | Mouse anti-105 (SN6) | ab11414 | Abcam | 1:500 | | | |
| 30 | Rabbit IgG1 polyclonal Isotype control | ab37415 | Abcam | 1:2000 | | | 1:2000 |
| 31 | Mouse IgG1 kappa monoclonal Isotype control | ab81032 | Abcam | 1:1600;1:500 | | | 1:500 |
| 32 | Goat anti-rabbit Alexa fluor 405,488, 568, 647 | A31556 A11034 A11011 A21245 | Thermo Scientific | 1:200 | 1:400 | | 1:200 |
| 33 | Goat anti-mouse Alexa fluor 405,488, 568, 647 | A31553 A28175 A11031 A21235 | Thermo Scientific | 1:200 | 1:400 | | 1:200 |

grown in 7H9 media (BD Difco) supplemented with 10% Albumin–Dextrose–Catalase (ADC, BD, Difco) and incubated in an orbitary shaker at 100 rpm, 37 °C until the mid-log phase. *GFP-H37Rv* was prepared by electroporating virulent *H37Rv* strain with *pMN437-GFPm2* vector (Addgene, 32362) and was maintained at 50 µg/ml hygromycin 7H9-ADC media. Single cell suspension required for carrying out infection experiments were prepared by passing bacterial cultures through a series of different gauge needles: five times through 23 and 26 ga and thrice through 30 ga.

All cell infection experiments (CFU, confocal, ELISA) were carried out at 1:10 Multiplicity of infection (MOI). For macrophage experiments, bacterial infection was set up for 4 h followed by RPMI wash and addition of amikacin sulfate at a final concentration of 200 µg/ml for 2 h to kill any extracellular bacteria. For ADSCs, infection was done for 12 h followed by addition of amikacin sulfate (200 µg/ml) for 2 h and replenishment of fresh media. All the treatments of drug, cytokines, or chemical inhibitors were done 24 h before the 6th day and 3rd day time point for ADSC and dTHP-1, respectively. For ADSCs siRNA transfections were done 48 h before the 6th day time point. Bacterial CFU was enumerated by adding lysis buffer (7H9 containing 0.1% SDS) in the required plate and incubating for 5 min and plating on 7H11 agar plates supplemented with OADC (BD Difco). The plates were incubated at 37 °C to allow bacterial growth, and counts were performed after 21 days. Drug-tolerant population was calculated as percent CFU in antibiotic-treated group with respect to the corresponding control group. For example, If CFU with IFNγ treatment is X and IFNγ + INH treatment is Y, % tolerant population would be Y/X × 100.

**MTT assay**. Cell viability was assessed by performing MTT (sigma) [3-(4,5-dimethyl-2-thiazolyl)-2,5-diphenyl-2H-tetrazolium bromide] assay. At indicated time points, media were removed from the plate and washed once with phenol free RMPI. MTT was prepared in phenol free RPMI at a working concentration of 1 mg/ml. 100 µl of MTT solution was added to each well of 96 well plates and incubated for appropriate time in cell incubator. Thereafter, MTT solution was removed and formazan crystals were dissolved in 100 µl DMSO. Samples obtained thereafter were quantified by measuring their absorbance at 560 nm in the plate reader.

**Immunoblotting**. For western blotting experiments, ADSCs were washed with ice cold PBS before their incubation with Buffer A solution (20 mM HEPES, 10 mM NaCl, 1.5 mM MgCl$_2$, 0.2 mM EDTA, and 0.5%v/v Trition-X-100) with 1X Protease Arrest (G−Biosciences) for 15 min on ice for lysis. Cell lysate was centrifuged at 4 °C at 6000 $g$ for 10 min and supernatant was collected. Protein quantification was done using BSA as standard in Bradford's assay. Protein sample was mixed with 6× loading dye and subjected to SDS PAGE and transferred to nitrocellulose membrane for immunoblotting. Blocking was done for an hour with Odyssey blocking buffer (LI-COR Biosciences) in 1:1 dilution with 1X PBS at the room temperature. Blots were immunoblotted with primary (1:1000) and then with secondary antibody (1:15000) made in blocking buffer. Blots were imaged with Odyssey Infra Red Imaging system (LI-COR Biosciences).

**Confocal microscopy**. For confocal microscopy experiments, bacteria were stained with PKH67, a green lipophilic dye, according to the manufactures protocol and resuspended into final media and incubated with cells for infection. To visualize acidified compartments, LysoTracker red dye (LysoTracker Red DND-99; Life Technologies) was added to the sample wells at a concentration of 500 nM for 30 min prior to fixation. Cells were fixed in 4% (w/v) PFA for 15–20 min, followed

by PBS wash twice. For antibody staining, cells were treated with ammonium chloride for 15 min. Cells were again washed with PBS and incubated with 0.2 % TritonX-100 in 1× PBS for 20 min to ensure permeabilization. It was followed by blocking solution (3% BSA in 1× PBST) for 1 h. Cells were then incubated with primary antibody (1:200) for 2–3 h at RT, followed by PBST wash and conjugated secondary antibody (1:400) for an hour. Cells were given a final wash with 1× PBS and coverslips were mounted in ProLong Gold antifade reagent (Life Technologies). Images were acquired by NIS-Elements software using the Nikon A1R laser scanning confocal microscope equipped with a Nikon Plan Apo VC ×20, NA 0.75, and Plan Apo VC ×100 oil, NA 1.40 objectives were used. Serial confocal sections, 0.5 μm thick, were acquired with a z-stack spanning 10–15 μm to form a composite image. Images were analysed using Imaris, NIS-Elements and image J software

**Flow cytometry**. Surface and intracellular protein staining in ADSCs were carried out using flow cytometry. At required time points, cells were scrapped off, pelleted, and washed. Cells were pelleted down at 1000 rpm and blocked in 3% BSA in 1× PBS and incubated with primary antibody (1:100) for 3 h in blocking buffer followed by incubation with alexa flour 488 conjugated secondary antibody (1:200) for 2 h (surface expression). Intracellular expression was assessed after permeabilizing cells with 0.05% saponin, followed by blocking, primary (1:100) and secondary antibody (1:200) incubation. After incubations cells were washed with 1× PBS and resuspended in 1× PBS and samples were acquired in BD FACS Canto II by using FACS Diva acquisition software. For measurement of cellular ROS, cells were scrapped at required time point and stained with CellROX Green before acquisition on BD FACS Canto II. Staining of the dyes were performed as per the manufacturer's directions. The data were analyzed using Flow Jo V10.5.3 software.

**Real time PCR and microarray**. Total RNA from ADSCs was isolated using mdi RNA isolation kit. cDNA was synthesized from 500 ng of total RNA by reverse transcriptase PCR using Bio-RAD iScript cDNA synthesis kit according to the manufacturer's protocol. The cDNA samples were run in triplicate using β-tubulin and actin as normalizing control, respectively, using SYBR green dye for real time fluorescence acquisition on the Bio-Rad CFX 96 Real time PCR system. Primers were custom synthesized from Sigma Aldrich Chemicals Ltd. Primers used: *ABCC1* (F:CGAGAACCAGAAGGCCTATTAC, R:ACAGGGCAGCAAACAGAA) *ABCG2* (F:CTTCGGCTTGCAACAACTATG,R:CCAGACACACCACGGATAAA), *Tubulin* (F:TTGGCCAGATCTTTAGACCAGACAAC,R: CCGTACCACATCCAGGACAG AATC),*Actin* (F: ACCTTCTACAATGAGCTGCG, R: CCTGGATAGCAACGTA CATGG).

For microarray, total RNA from ADSC uninfected or infected H37Rv was extracted using mdi RNA isolation kit. Samples were sent to Bionivid Technologies, Bangalore for cDNA synthesis and hybridization to 25 "Illumina human WholeGenome-6 version 2 BeadChips" using standard Illumina protocols. Six samples (three replicates each for uninfected and infected cells) were used for hybridization. The raw files for the microarray experiment are available at GEO database "GSE133803". Microarray data were analyzed using R packages (GenomeStudio, beadarray, genefilter, limma).

**C57BL/6 aerosol challenge**. All mice experiments were carried out in the Tuberculosis Aerosol Challenge Facility (TACF, ICGEB, New Delhi, India). C57BL/6 mice were housed in individual ventilated cages contained within the biosafety level 3 enclosure. The animal holding area of the BSL3 lab is maintained at 20–25 °C, 30–60% humidity and 12–12 h of light–dark cycle. Aerosol challenge of ~100 CFUs was given to the animals in a Wisconsin–Madison chamber according to the standardized protocol in the TACF facility. To check for infection establishment, two animals were selected randomly and humanely euthanized 24 h postaerosol challenge. The lungs and spleen tissues were harvested and homogenized to enumerate CFU. Tissue lysates were serially diluted and plated on petri plates containing Middlebrook 7H11 agar (Difco) supplemented with 10% OADC (Becton, Dickinson) and 0.5% glycerol.

**Animal dosing, CFU plating**. For in vivo experiments, drug dosing was initiated 4 weeks postaerosol challenge and the animals were administered the drugs by oral gavage at a dose of 10 mg/kg INH and 50 mg/kg celecoxib in combination or individually every day till additional 8 weeks. After 4 and 8 weeks post treatment, equal number of mice in each group were euthanized, followed by removal of the lung and spleen which were homogenized followed by plating on 7H11-OADC plates.

*FACS Sorting, CFU plating*: Half lung was used in the preparation of single cell suspension followed by FACS sorting and subsequent plating. To begin with the tissue was washed with PBS, chopped in to small pieces followed by addition of 20U/ml DNAses, 1 mg/ml collagenase D and incubated for 30 min at 37 °C. Cells were passed through nylon mesh to get single cell suspension. The cells were pelleted at 2000 rpm, treated with RBC lysis buffer followed by washing with PBS. Cells were stained with the antibody cocktail of CD45-APC (clone 30-F11; 559864), CD73-BV450 (clone TY/23; 561544), CD11b-APC-Cy7 (clone M1/70; 557657), Ly6G-PE (clone 1A8; 551461), Sca-1-PE-CF594 (clone D7; 562730) and just before sorting PI was added at 5 μg/ml for live/dead staining. All the cells from half of the

lung were sorted to get the maximum number of sorted cells which were then pelleted, lysed and plated for getting bacterial CFU.

**Tissue Immunohistochemistry**. Five-micron thick sections of formalin-fixed paraffin-embedded tissues were taken on the coated slide. Deparaffinization was done by dipping the slides in xylene for 5 min (two changes), acetone for 2–3 min, alcohol for 2–3 min and then under running/tap water. Antigen retrieval was performed with citrate buffer (pH 6) in microwave oven, at 100 °C, 900 MW for 30 min. Samples were allowed to cool down, washed thrice with Tris buffer (pH 7.5). Endogenous peroxidase blocking was done with 4% Hydrogen peroxide in methanol for 20 min. Rabbit anti-Ag85B (ab43019) primary antibody (pH 6) was added (1:2000) and incubated for 2 h at the room temperature. Thereafter, three washings were given with Tris buffer (pH 7.5). Universal polymer-based secondary antibody (Skytek Laboratories, USA) was incubated at the room temperature for 30 min, and the reaction product was developed with 3,3′-diaminobenzidine chromogen. Color development was monitored under the microscope. Skin biopsy from patients with cutaneous tuberculosis was used as positive control. Three subsequent washings with Tris buffer were given at 5 min interval. After that the Mouse anti-CD73 antibody (ab91086, pH 6, dilution 1: 500) was added and incubated in the room temperature for overnight. Three washings were given with TRIS buffer. Alkaline phosphatase tagged goat anti-mouse IgG H&L secondary antibody (Vector, MP5402) was added for 30 min at the room temperature. Three washings were given in Tris buffer. A VECTOR® Blue Alkaline Phosphatase chromogen was used to develop the color of the reaction (Blue AP), prepared in Tris HCL (pH 8.2) and the color development was monitored under microscope. The slides then were washed under running tap water for 3–5 min, counterstained with Fast Red stain for 5 s and air-dried before mounting with a glycerol solution. Along with the positive controls, isotype controls were used in same batch while staining, as follows: Rabbit IgG polyclonal isotype control (ab 37415) was added in 1:2000 dilutions in the place of anti-Ag85B and incubated for 2 h the at room temperature. Rest of the steps was same as described above including the secondary antibody. In the place of anti-CD73 antibody, Mouse IgG1, kappa monoclonal (NCG01) isotype control was (ab81032) used at 1:500 dilution. Rest of the steps followed are same as above.

**Tissue Immunofluorescence staining**. Dual immunofluorescence study was performed on formalin-fixed paraffin-embedded (FFPE) lung biopsies. The sections cut were deparaffinized in xylene for 5 min (two changes), acetone for 2–3 min, alcohol for 2–3 min and then under running/tap water. Thereafter the slides were washed in RO water for three times and antigen retrieval was done in pH 9 for 40 min at the room temperature. Thereafter the sections were treated with Proteinase K (Sigma, dilution 1: 25) for 30 min. The slides were then put in Tris EDTA buffer at pH 9 at 4 °C for 40 min, followed by three washings in Tris buffer for 5 min each. Rabbit anti-Ag85B (ab43019) primary antibody (pH 6) was added (1:2000) and incubated for 2 h at the room temperature. Thereafter Goat anti-Rabbit IgG (H + L) highly cross absorbed secondary antibody conjugated with Alexa Fluor 488 (Thermofisher, dilution: 1:200) was incubated at room temperature for 15 min and observed under fluorescent microscope green channel. The slides were washed in Tris buffer for three times 5 min each. Subsequently the Mouse anti-CD73 antibody (ab91086, dilution 1: 1600) or anti-CD105 antibody (SN6, ab11414, dilution 1: 500) was added and incubated in the room temperature for about 2.5 h. Thereafter Goat anti-mouse IgG (H + L) highly cross absorbed secondary antibody conjugated with Alexa Fluor 568 (Thermofisher, dilution: 1:200) was incubated at the room temperature for 15 min and observed under fluorescent microscope red channel. The Ag85B stained organisms were identified by green fluorescence and the CD73 and CD105 positive stromal cells were identified by red fluorescence. Thereafter the slides were washed with RO water for two times and mounted with glycerin and stored in 4 °C. All steps were performed in dark room. After satisfying regarding the staining the colocalization analysis was done with confocal microscope as described earlier. Skin biopsy from patients infected with *Mtb* was used as positive control. Along with the positive controls isotype controls were used as follows: Rabbit IgG polyclonal isotype control (ab 37415) was added in 1:2000 dilutions in the place of anti-Ag85B and incubated for 2 h at the room temperature. Rest of the steps were same as described above including the secondary antibody. In the place of CD73 and CD105 antibodies, Mouse IgG1, kappa monoclonal (NCG01) isotype control was (ab81032) used in 1:1600 and 1:500 dilutions, respectively. Rests of the steps followed are same as above.

**Statistical analysis**. Data compilation was done using Microsoft Excel (2011). Statistical significance for comparisons between two sets of the experiments was done using unpaired two-tailed Student's $t$ test. For multiple treatment experiments, one-way ANOVA followed by multiple comparison analysis was performed in GraphPad PRISM 8. For animal experiments, nonparametric Mann–Whitney $U$ test was performed.

**Reporting summary**. Further information on research design is available in the Nature Research Reporting Summary linked to this article.

## Data availability

The raw files for the microarray experiment are available at GEO database "GSE133803 [https://www.ncbi.nlm.nih.gov/geo/query/acc.cgi?acc=GSE133803]". For all the plots in Figs. 1, 2, 3, 4 and Supplementary Figs. 1, 2, 3, 4, 5, 6, 7b, corresponding raw data are provided in the Source Data File. Any additional relevant data are available on request. Source data are provided with this paper.

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

## Acknowledgements

The research in D.K.'s group is supported by Wellcome-DBT India Alliance Senior fellowship (IA/S/17/1/503071). This work was partially supported by DBT, Govt. of India (BT/PR14730/BRB/10/874/2010) and SERB (EMR/2016/005296). N.J and V.S. are recipients of SRF from CSIR, India and H.K. has received SRF from UGC. We thank Sanal MG for useful inputs and Aditya Rathee for help with FACS sorting. All the work involving *Mycobacterium tuberculosis* including animal infection experiments were performed at Tuberculosis Aerosol Challenge Facility (TACF), a DBT sponsored national facility hosted at ICGEB campus.

## Author contributions

Concept and experiment design: N.J., D.K.; experiments: N.J.; animal experiments: N.J., L.S., V.S.; data analysis: N.J., H.K., P.D., V.A., S.K., D.K.; IHC and IFA experiments on human samples: P.D., S.K., V.A., N.J.; manuscript writing: N.J., P.D., D.K.; funding: D.K.; overall supervision: D.K. All authors reviewed and commented on the manuscript and agreed to the final version.

## Competing interests

The authors declare no competing interest.
