## [Peer Review File · Nature Communications]

Reviewers' Comments:

Reviewer #1:

Remarks to the Author:

Summary: This paper investigates mechanisms by which mesenchymal stem cells protect M.tb bacilli from antibiotics (transporter pumps) and immune responses (variety of different/unusual responses to proinflammatory cytokines that traditionally activate macrophages but have different effects on MSCs). The research tools were primarily in vitro using human adipose-derived stem cells with additional supporting information from mouse studies and tissues from TB patients. The overall findings will be of substantial interest to the TB research and clinical community, as the results provide insight and a direction to develop and screen/test therapies that may eliminate M.tb from this cell population. Statistical analyses were primarily 2-tailed student's t-tests (which seems OK) but possibly the authors should consider if other tests are appropriate as well, for example ANOVAs?

Major Critiques: This is a solid paper with experimental evidence supporting the conclusions. My main critique is whether experimental results were reproducible in Fig 3 and 4. Please report how many times the experiments were performed. As written, it seems like experiments were performed once with 1 lot of cells throughout or 1 set of animals. Twice would be minimum, and three times better. If experiments were performed once, please repeat the most critical set.

Minor Critiques:

1. Recommend an editor to find and correct the minor language, grammar mistakes. A few word choices are non-standard.
2. Double check effect versus affect.
3. I may have missed it in the methods. Please clarify how you defined, identified and calculated "tolerance" on the Y-axes of many graphs.
4. Suggestions for data presentations: For the mouse studies - consider putting CFUs on a linear scale. I know this is unusual but changing the scale will better show the differences that are lost on a Log10 transformation. For the IHCs and photomicrographs, consider higher magnification. It is too hard to see what the arrows are pointing to.
5. Considering adding to Intro or discussion additional reference on MSCs by last author Campos-Neto in AJP.

Reviewer #2:

Remarks to the Author:

In the current study Jain et al. address the Mtb infection of mesenchymal stem cells (MSC) in vitro and in vivo. As MSCs have only recently identified and been proposed to play an important role in during infection. The detailed functional role of MSC in the context of infection however is not known. Thus the authors address a highly relevant issue very likely having an impact on TB vaccination and also TB therapy strategies.

The authors present a very thorough and very detailed analysis of the infection of MSC by Mtb in vitro and in vivo. The data are very convincing and the experiments are well controlled. The paper is well written and provides novel and thought provoking mechanistic insights into the role of MSC in the Mtb infected host. However a few things should be carefully considered, as the authors state in their

discussion the limited reliance of ex vivo data. Said that the authors should readdress the histological findings in TB patients.

- Fig4 H: The IHC data provided in Figure 4H look specific. However the respective controls should be provided by the authors in the supplementary part.
- Fig4 I: In contrast to Fig4H, Fig4I is not convincing. First larger orange arrows lie on top of smaller arrows, if you zoom into the picture. This is misleading and should be avoided. Irrespective of the arrows the CD73 positive cells cannot be identified with the blue counterstaining within the pictures of the tissue sections provided. This needs to be readdressed in detail.
- The authors do state in their text that Ag85 and CD73 are "in close vicinity" but this is in fact several cell diameters away. It appears that in contrast to their B16 mouse data, the authors do not see CD73 cells harbouring Mtb bacteria in human TB lesions. The apparent key question is: Do MSC in the human TB patient harbour Mtb bacteria as the mouse data suggest, or does that differ. In order to address this the authors may consider e.g. a version of a proximity ligation assays (e.g. CD73 vs Ag85) to demonstrate the presence of Mtb within MSC or may use an independent staining protocol, where the signals can be undoubtedly visualized and clearly separated from each other. Another way to address this may be the use of a Ziehl-Neelsen-stain followed by IHC for CD73.
- The wording used by the authors should be more cautious in some instances. The authors should refrain from exaggerating and overstating their findings: e.g. l.121: please use "(strongly) suggest" or similar instead of "evident", or l.144: "reduced" instead of "wiped out" because 2-4% are still left. As mentioned above (l.341), the authors do state that Ag85 and CD73 are "in close vicinity" but this is in fact several cell diameters away. However this point needs to be addressed in further experiments anyway. Please check the whole manuscript for these issues, since a more precise wording would improve the quality of the manuscript.

Reviewer #3:

Remarks to the Author:

Comments to the Author

In the manuscript, "Mesenchymal stem cells (MSCs) offer a drug-tolerant and immune privileged niche to Mycobacterium tuberculosis" the authors present a study where they show that MSC provide a favorable environment for Mtb to reside and becomes more drug resistant. They hypothesize that there is a role for ABC transporters in drug tolerance and they find that lysosomal function rather than efflux activity is responsible to achieve the drug-tolerant phenotype. Authors also show a pro bacterial roles for inflammatory cytokines and lipid mediator PGE2 in MSC. The authors showed the novel ability of MSCs to provide a niche for Mtb to survive in harsh conditions.

The manuscript is overall interesting, however, the data needs improvement before publication.

Introduction:

The introduction is nicely written, but authors could provide a little information on why they have used and adipose-derived stem cells specifically.

General comments:

All supplementary figures lack statistical analysis

Specific comments

Results:

1st part: Adipose-derived Mesenchymal stem cells (ADSCs) support mycobacterial growth

- Authors have characterized ADSCs by using just two markers CD73 and CD271 markers. However, CD73 marker is not specific for ADSCs, it is also present in lymphocytes and many of the cancerous cell lines. It would be useful to use additional markers like CD90, CD44 or CD105 along with a dump channel marker like CD11b.
- Also, in figure S1A, authors have shown two separate histograms for CD73 and CD271, I would suggest showing a bivariate plot and the gating strategy.
- For characterization of differentiation of ADSCs into chondrocytes, adipocytes, and osteocytes, authors could also perform a flow cytometry along with staining to confirm the results.
- Line 88, fig 1B, authors shown only day 6. Other time points showing the kinetics of growth would be really informative showing MFI ranging from lowest to highest. It will provide readers a better idea to whether the bacteria are multiplying inside or not.
- In fig 1C-F, please convert the CFUs to log scale and make the same scale for all plots. Or at least use the same scale so it is easier to visualize. Additionally, please show CFUS up to day 12 for all conditions and infections. This will make it easier to compare the four graphs.
- Fig S1C, it's the same as point 2, authors can do an additional flow cytometry or any other analysis to confirm that ADSCs do not differentiate after H37Rv infection. Also, the authors have not mentioned what is the right lane in that image. They could write legend on the top.

2nd part: ADSC resident Mycobacterium tuberculosis shows drug-tolerant phenotype

- Why authors have selected day 9 to plate CFUs? It would be good to show several timepoints and explain why they chose 9 days based on results.
- Authors have compared the treatment of INH and Rif on ADSCs with THP-1. I understand that ADSCs used here are primary cells and THP-1 is a monocytic cell line. The comparison would be more realistic they use primary macrophages (minor).
- Fig 1H, 5 ug.ml INH has the least CFUs, where are the plots for that concentration.

3rd part: Host ABC transporters ABCC1 and ABCG2 play a key role in bacterial drug tolerance

- Microarray data could be presented in a way where authors can highlight and show upregulation and downregulation of different genes. And they can point out the specific ones.
- Fig 1J, I am not convinced of the upregulations of ABCC1 and ABCG2 based on those histograms, maybe if they write down the percentage of increase in ABCC1 and ABCG2 receptors? It really does not look to be statistically significant.
- Line 131, authors have not mentioned the timeline and dose of H37Rv infection, cell harvesting, and staining
- Supplementary figure S2G and H, Please show the same scale in y-axis

4th part: Role of lysosomal function in mycobacterial drug tolerance in ADSCs

- Results still do not provide a valid justification of fig 1, why non-drug resistant Mtb is increasing after inhibition of ABCC1 and ABCG2. A better explanation of why BafA1, 3MA, and Chloroquine inhibition increase CFUs of non-drug resistant Mtb. It is still confusing.

5th part: Effect of inflammatory cytokines IFN γ and TNF α on drug tolerance within ADSCs

- Line 175, Could you please provide more references for this statement?
- Line 180, Is it figure 2D?
- Fig 2E and 2F, why concentration and units of measurement are different? If this is the case, then we cannot compare between these two (Line 184).
- Line 193-194, How can we say that Mtb is rescued from cytokine-mediated killing?

6th part: Analysis of intracellular niches of Mtb shows classic phagosome maturation dynamics in ADSCs

- Fig S3A, please mention the scale and magnification.
 - Fig 2K, Line 215-216, It is mentioned that BCG is present mainly in Lysotracker-LAMP1 double positive compartment, but from the figure, it seems like the highest percentage of BCG is in Lysotracker compartment.
 - Line 223, Fig S2E, For ABCC1, Mtb seems to colocalize but I am not convinced with ABCG2.
 - Line 226-227, How it is giving an impression that ABCC1 is directly involved in killing?
- 7th part: The lipid mediator PGE2 helps MSCs exhibit pro-bacterial attributes
- Figure S5C, actually shows the opposite of what the authors mention in 272. Celecoxib reduces CFUs in THP1 30% while only 10-15% in MSCs
- 8th part: MSCs serve as a niche for Mtb during in vivo infection allowing drug tolerance in PGE2 dependent manner

- Please show the group with the aerosol challenge (control)
- When INH+ Celecoxib were given in combination, where the doses of these two 50 and 10mg/kg respectively?
- Fig 4A, line 300-301, it is not clear from the figure that there are differences in CFUs, maybe if authors present the figure in log scale, things could be clear.
- Plot fig 4A and 4B please show the figures with the scale same.
- Fig 4C, I am not convinced with the markers used for characterizing MSCs and macrophages populations. I would suggest using more markers like CD44+, CD90+, CD105+ and CD45- and CD11B- for MSCs and Ly6C+ or F4/80+, CD11C-, Ly6G- for macrophages.
- Line 136, Fig 4F, how did you count cells?
- Fig 4G, please explain why CFU can decrease at week 8 and then increase again in case of the group treated with celecoxib?

Materials and methods:

- Reagents: Please provide all the necessary information about the antibodies like the fluorophore attached, clone etc. Also, few of the antibodies have not been written, please add all.
- Please explain where did the H37Rv came from or how it was transfected
- Please mention the strain of BCG used in this study.
- Why authors have used ADC and not OADC?
- Line 530, Why 4 hours for macrophage and 12 hours for ADSCs?
- At what time were plates counted?
- Which version of FlowJo was used?
- How many reads were acquired in Illumina?
- Please mention how mice were euthanized.
- Tissue processing, it seems like authors have processed lung and spleen in the same way, which is probably not the case. Please write the protocol clearly
- Please mention the dilution used for anti-rabbit IgG antibody (Line 634).

Marcela Henao-Tamayo

Reviewers' comments:

Reviewer #1 (Remarks to the Author):

Summary: This paper investigates mechanisms by which mesenchymal stem cells protect M.tb bacilli from antibiotics (transporter pumps) and immune responses (variety of different/unusual responses to proinflammatory cytokines that traditionally activate macrophages but have different effects on MSCs). The research tools were primarily in vitro using human adipose-derived stem cells with additional supporting information from mouse studies and tissues from TB patients. The overall findings will be of substantial interest to the TB research and clinical community, as the results provide insight and a direction to develop and screen/test therapies that may eliminate M.tb from this cell population. Statistical analyses were primarily 2-tailed student's t-tests (which seems OK) but possibly the authors should consider if other tests are appropriate as well, for example ANOVAs?

Thank you for appreciating this study and recognizing its biomedical relevance. In the revised manuscript, we have performed additional statistical tests like ANOVA and rank-based tests wherever appropriate.

Major Critiques: This is a solid paper with experimental evidence supporting the conclusions. My main critique is whether experimental results were reproducible in Fig 3 and 4. Please report how many times the experiments were performed. As written, it seems like experiments were performed once with 1 lot of cells throughout or 1 set of animals. Twice would be minimum, and three times better. If experiments were performed once, please repeat the most critical set.

Thank you for this very critical observation. We acknowledge that the data presented in Fig. 3 in

Figure 3 (A) Uninfected or H37Rv infected ADSCs were either untreated or treated with IFN γ (5 ng/ml), TNF α (20 ng/ml), MK571 (50 μ M) and celecoxib (250 μ M) for 24 hours before performing supernatant ELISA on 6th day post-infection (post-seeding in uninfected cells). Data is shown for 2 independent donars (blue, orange dots) (B) H37Rv infected ADSCs were treated with different doses of celecoxib (50 μ M, 150 μ M, 250 μ M) in addition to IFN γ (5 ng/ml), TNF α (20 ng/ml) or MK571 (50 μ M) for 24 hours before the cells were harvested for CFU plating on 6th day post-infection. Again, data is shown for 3 independent donars (blue, green and purple dots)

the original manuscript were from ADSCs obtained from only one donor, while we did multiple experiments for each of the observations. In view of the suggestion made by this reviewer, we have now performed experiments with ADSCs isolated from two additional and separate donors. This is true for Fig. 3A, 3B and 3E. For Fig. 3A and Fig. 3B, results from separate cell lots are also shown in the revised figures.

Similarly, for results in Fig. 4, we have now repeated the same experiment two times (making a total of three independent experiments). The results in Fig. 4 are revised accordingly.

Thus the results reported in ADSCs and in the mice are indeed reproducible.

Minor Critiques:

1. Recommend an editor to find and correct the minor language, grammar mistakes. A few word choices are non-standard.

We have tried to address these concerns of the reviewer in the revised manuscript, taking help from "Grammarly".

2. Double check effect versus affect.

Thank You, we have tried to address this in the revised version.

3. I may have missed it in the methods. Please clarify how you defined, identified and calculated "tolerance" on the Y-axes of many graphs.

We have further explained the calculation of drug tolerance in the revised methods section as follows:

Page 25, Line 613: "Drug tolerant population was calculated as percent CFU in antibiotic-treated group with respect to the corresponding control group. For example, If CFU with IFN γ treatment is X and IFN γ +INH treatment is Y, % INH tolerant population would be $Y/X*100$."

4. Suggestions for data presentations: For the mouse studies - consider putting CFUs on a linear scale. I know this is unusual but changing the scale will better show the differences that are lost on a Log₁₀ transformation. For the IHCs and photomicrographs, consider higher magnification. It is too hard to see what the arrows are pointing to.

We agree with the reviewer that at least in few of the CFU plots, differences would look better on a linear scale. However with two additional animal experiments, we have noted more refined differences between CFUs and therefore we have stuck to representing the data in the conventional Log₁₀ scale. For IHC, we have repeated and further standardized the dual staining of granuloma from human lung biopsy samples. We are pleased to state that we could achieve considerable improvisation in the histopathology images. These new images are added into the revised manuscript.

5. Considering adding to Intro or discussion additional reference on MSCs by last author Campos-Neto in AJP.

Thank you for this suggestion, we have included this reference.

Reviewer #2 (Remarks to the Author):

In the current study Jain et al. address the Mtb infection of mesenchymal stem cells (MSC) in vitro and in vivo. As MSCs have only recently identified and been proposed to play an important role in during infection. The detailed functional role of MSC in the context of infection however is not known. Thus the authors address a highly relevant issue very likely having an impact on TB vaccination and also TB therapy strategies.

We are grateful to this reviewer for appreciating our work and recognizing its relevance to tuberculosis prevention and therapy.

The authors present a very thorough and very detailed analysis of the infection of MSC by Mtb in vitro and in vivo. The data are very convincing and the experiments are well controlled. The paper is well written and provides novel and thought provoking mechanistic insights into the role of MSC in the Mtb infected host. However a few things should be carefully considered, as the authors state in their discussion the limited reliance of ex vivo data. Said that the authors should readdress the histological findings in TB patients.

We again thank the reviewer for clearly articulating the strengths of our manuscript as well as the concerns related to ex vivo data and histological data.

- Fig4 H: The IHC data provided in Figure 4H look specific. However the respective controls should be provided by the authors in the supplementary part.

As suggested by the reviewer, we have included the controls for the histopathology data in the revised manuscript as Fig. S7C.

- Fig4 I: In contrast to Fig4H, Fig4I is not convincing. First larger orange arrows lie on top of smaller arrows, if you zoom into the picture. This is misleading and should be avoided. Irrespective of the arrows the CD73 positive cells cannot be identified with the blue counterstaining within the pictures of the tissue sections provided. This needs to readdressed in detail.

We have replaced figures 4H and 4I with a much higher resolution images showing clear distinction between the dual staining. The arrows are shown prominently and we believe the histopathology data in the revised manuscript matches up to the desired standard.

Figure S7C For antibody controls, skin biopsy from a patient with known *lupus vulgaris* infection was included. Photomicrograph (a) shows dual *anti-Mycobacterium Ag85B antigen* positivity in the bacilli (brown) and anti-CD73 stain (red) colocalization in the mesenchymal cells scattered around the epithelioid granulomas [a x 400]. Photomicrograph (b) shows secondary only control processed with anti-CD73 staining [b x 200]. Figure (c) shows dual *anti-Mycobacterium Ag85B antigen* (brown) and anti-CD105 stain (red) colocalization in the same cells described above [a x 400]. The *anti-Mycobacterial Ag85B stain* is highlighting the fragmented bacillary form, rounded coccoid forms and cytoplasmic granular antigens of the organism. Figure (d) is the secondary only control of the same biopsy tissue processed with CD105 staining [d x 100].

- The authors do state in their text that Ag85 and CD73 are “in close vicinity” but this is in fact several cell diameters away. It appears that in contrast to their BL6 mouse data, the authors do not see CD73 cells harboring Mtb bacteria in human TB lesions. The apparent key question is: Do MSC in the human TB patient harbor Mtb bacteria as the mouse data suggest, or does that differ. In order to address this the authors may consider e.g. a version of a proximity ligation assays (e.g. CD73 vs Ag85) to demonstrate the presence of Mtb within MSC or may use an independent staining protocol, where the signals can be undoubtedly visualized and clearly separated from each other. Another way to address this may be the use of a Ziehl-Neelsen-stain followed by IHC for CD73.

Previously, the dual staining in human granuloma samples were not as well standardized and therefore we could not really see their co-localization. In the revised manuscript we indeed show clear co-localization between CD73 and Mtb. In addition, we also observe similar co-localization between CD105 (another MSC marker) and Mtb. Therefore we have firmed up our inference here and instead of saying “in close vicinity”, we do mention “MSC harboring Mtb could be seen in and around the granulomas”.

Figure 5(B) Photomicrographs showing trucut biopsies of lung parenchyma from patients with pulmonary tuberculosis. Confluent epithelioid cell granulomas with Langhan’s giant cells (black arrows) and areas of necrosis (pink arrow) were seen [H&E stained slides: a x 100; b x 100; c x 200]. Tissue sections were stained with dual anti-Mycobacterial Ag85B (brown chromagen) and anti-CD73 (red chromagen) antibodies. The epithelioid cell cytoplasm forming the core of the granulomas showed positivity for Mycobacterium tuberculosis antigen (black arrows) and there was polarization of the red stain positive mesenchymal cells towards the granulomas (blue arrows). Some of the epithelioid cells and mesenchymal cells were positive for both Mycobacterium tuberculosis antigen (black arrows) and CD73 antigens (blue arrows) [d x100; e x200; fx200]. With the dual anti-Mycobacterial Ag85B stain and anti- CD105 antibodies also similar pattern of Mycobacterium tuberculosis antigen positivity (black arrows) was noted inside the granulomas and Langhan’s giant cell, surrounded by the CD105 antibody positive mesenchymal cells (blue arrows). Many of the mesenchymal cells were found to have colocalization of both the Mycobacterium tuberculosis antigen and CD105 stains (marked by both black and blue arrows) [g x100; h x200; i x400]. A small area from d (green arrow) has been zoomed and shown as inset (green) marking co-localization of CD73 and Ag85B. The inset in i (green) is drawn to highlight the co-staining.

- The wording used by the authors should be more cautious in some instances. The authors should refrain from exaggerating and overstating their findings: e.g. l.121: please use "(strongly) suggest" or similar instead of "evident", or l.144: "reduced" instead of "wiped out" because 2-4% are still left. As mentioned above (l.341), the authors do state that Ag85 and CD73 are "in close vicinity" but this is in fact several cell diameters away. However this point needs to be addressed in further experiments anyway. Please check the whole manuscript for these issues, since a more precise wording would improve the quality of the manuscript.

Thank you for the suggestions and we have accordingly tried to address these concerns throughout the manuscript.

Reviewer #3 (Remarks to the Author):

Comments to the Author

In the manuscript, "Mesenchymal stem cells (MSCs) offer a drug-tolerant and immune privileged niche to Mycobacterium tuberculosis" the authors present a study where they show that MSC provide a favorable environment for Mtb to reside and becomes more drug resistant. They hypothesize that there is a role for ABC transporters in drug tolerance and they find that lysosomal function rather than efflux activity is responsible to achieve the drug-tolerant phenotype. Authors also show a pro bacterial roles for inflammatory cytokines and lipid mediator PGE2 in MSC.

The authors showed the novel ability of MSCs to provide a niche for Mtb to survive in harsh conditions.

The manuscript is overall interesting, however, the data needs improvement before publication.

Thank you for finding our manuscript of interest and clearly articulating how to further improve the study.

Introduction:

The introduction is nicely written, but authors could provide a little information on why they have used and adipose-derived stem cells specifically.

MSCs are present throughout the body however bone marrow (for animals) and adipose tissues (for humans) are among the easiest source to obtain them. Since the study was majorly planned with human cells, we used ADSCs. We have added little more description on why ADSCs were used in this study.

Page 4, Line 76: "MSCs can be readily isolated from bone-marrow (animals) and adipose tissues (humans) thereby serving as an excellent *ex vivo* model to study mycobacterial lifestyle in these cells."

General comments:

All supplementary figures lack statistical analysis

We have added statistical analysis to all the supplementary figures.

Specific comments

Results:

1st part: Adipose-derived Mesenchymal stem cells (ADSCs) support mycobacterial growth

- Authors have characterized ADSCs by using just two markers CD73 and CD271 markers. However, CD73 marker is not specific for ADSCs, it is also present in lymphocytes and many of

the cancerous cell lines. It would be useful to use additional markers like CD90, CD44 or CD105 along with a dump channel marker like CD11b.

We agree with the reviewer's concern. To address this concern, we have now characterized human ADSCs using additional markers as suggested, specifically CD90, CD44, CD105 and negative marker CD11b. These characterizations are added in the revised manuscript as figure S1A and S1B.

Figure S1(A) Line histogram of ADSCs stained with CD73, CD44, CD105, CD90 and CD271 (blue histograms) and corresponding isotype controls (Red histograms). (B) Dot plot of ADSCs costained with CD73-FITC and CD271-Alexa fluor 647, only Alexa fluor 647 - secondary control and CD73-FITC with CD11b-PE.

• Also, in figure S1A, authors have shown two separate histograms for CD73 and CD271, I would suggest showing a bivariate plot and the gating strategy.

We have included the dual staining for CD73 and CD271 in the revised manuscript (Fig. S1B; also shown above).

• For characterization of differentiation of ADSCs into chondrocytes, adipocytes, and osteocytes, authors could also perform a flow cytometry along with staining to confirm the results.

We could perform flow cytometry easily for adipocyte. For both chondrocytes and osteocytes, instead of performing flow cytometry we have performed RT-PCR for pathway specific genes to show the differentiation, which is routinely used¹. The revised results are included in the

supplementary figure S1C, D and E. We have also combined these assays for uninfected and H37Rv-infected ADSCs in this figure.

Figure S1(C) Uninfected ADSCs differentiated into osteocytes, using specific differentiation media according to the manufacturer's protocol and stained with safranin O. Cells were visualized under light microscope at 10X. These differentiated cells were analyzed for osteocyte lineage specific genes using real time PCR. Similarly, 6th day GFP- H37Rv infected ADSCs were stained with safranin O and visualized under fluorescence and light microscope. Parallely, they were processed for same specific genes for RT-PCR. (D) Uninfected ADSCs differentiated into chondrocytes, using specific differentiation media according to the manufacturer's protocol and stained with Alizarin Red S. Cells were visualized under light microscope at 10X. These differentiated cells were analyzed for chondrocyte lineage specific genes using real time PCR. Similarly staining and RT-PCR was carried out in 6th day GFP- H37Rv infected ADSCs. (E) Uninfected ADSCs were differentiated into Adipocytes, using specific differentiation media according to the manufacturer's protocol and stained with Oil Red O. Cells were visualized under light microscope at 40X. Additionally, undifferentiated (Blue histogram) and differentiated (Orange histogram) cells were also analyzed for lipid accumulation using LipidTox staining by flow cytometry. Oil Red O staining and LipidTox staining were also carried out in 6th day GFP- H37Rv infected ADSCs. Blue histogram represent uninfected ADSC while orange histogram represent 6th day infected ADSCs. Experiments were done in triplicate. Error bars represents S.D.

• Line 88, fig 1B, authors shown only day 6. Other time points showing the kinetics of growth would be really informative showing MFI ranging from lowest to highest. It will provide readers a better idea to whether the bacteria are multiplying inside or not.

In view of this observation by the reviewer, we have performed fresh experiments to calculate MFI across the time course of experiments. The images and corresponding quantified data sets are shown in Fig. 1A and 1B in the revised manuscript.

Figure 1(A) ADSCs were infected with GFP-Rv at 1:10 MOI, representative confocal images at 0 hour, 3rd day, 6th day, 9th day and 12th day post infection are shown. (B) MFI of GFP-Rv per cell across above mentioned time points were calculated for 5 fields repeated in more than triplicate.

- In fig 1C-F, please convert the CFUs to log scale and make the same scale for all plots. Or at least use the same scale so it is easier to visualize. Additionally, please show CFUS up to day 12 for all conditions and infections. This will make it easier to compare the four graphs.

We have revised the figure with same scale making the comparison easy between all the CFU results shown in figure 1.

Figure 1(C) Growth kinetics of H37Rv within ADSCs across 12 days post-infection. (D) Growth kinetics of H37Rv within human MDMs across 7 days post infection and (E) in THP-1 macrophages across 3 days. (F) Growth kinetics of vaccine strain BCG within ADSCs and THP-1 macrophages (G) across 0, 1, 2 and 3 days post-infection.

Moreover, while we would want to include up to Day 12 data for H37Rv in macrophages and BCG data in ADSCs and macrophages, it is difficult since macrophages show considerable cell death beyond Day 7 (MDM) and Day 3 (THP-1) post-infection. For BCG, post three days of infection, we could not get any CFUs in ADSCs and therefore we could not include any additional time points.

- Fig S1C, it's the same as point 2, authors can do an additional flow cytometry or any other analysis to confirm that ADSCs do not differentiate after H37Rv infection. Also, the authors have not mentioned what is the right lane in that image. They could write legend on the top. In addition to staining, we have performed flow cytometry for adipocytes and RT-PCR for chondrocytes and osteocytes. We have appropriately revised the legends in the revised manuscript. The results are included as Fig. S1C, D and E (also shown above).

2nd part: ADSC resident Mycobacterium tuberculosis shows drug-tolerant phenotype

- Why authors have selected day 9 to plate CFUs? It would be good to show several timepoints and explain why they chose 9 days based on results.

In the revised manuscript, we show all the relevant time points for ADSCs and MDMs.

Figure 1(G) H37Rv infected ADSCs were treated with doses of isoniazid (INH), and rifampicin (RIF) (0.1 – 5 µg/ml) for 24 hours prior to different time points of 3rd day, 6th day, 9th day and 12th day, and were plated for CFU enumeration. (H) Percent drug tolerant bacterial population to INH (1 µg/ml) within infected ADSCs (blue) on 3rd, 6th, 9th, 12th day and MDMs (purple) on 3rd, 5th and 7th day post-infection, respectively. (I) Percent drug tolerant bacterial population to RIF (0.5 µg/ml) within infected ADSCs (blue) on 3rd, 6th, 9th and 12th day and within MDMs (purple) on 3rd, 5th and 12th day post-infection, respectively.

- Authors have compared the treatment of INH and Rif on ADSCs with THP-1. I understand that ADSCs used here are primary cells and THP-1 is a monocytic cell line. The comparison would be more realistic they use primary macrophages (minor).

We have re-performed these experiments with human MDMs and now have replaced the entire drug-tolerance results involving THP-1 cells with those from human MDMs in the revised manuscript (Fig. 1G).

- Fig 1H, 5 µg/ml INH has the least CFUs, where are the plots for that concentration. While we have tolerant population data for each time point and doses, we only showed data from 1 µg/ml dose for INH throughout this study since that is the most relevant concentration used in the literature. In the revised manuscript, we have included tolerance data for all the doses tested as supplementary figure S2B.

Figure S2(B) Percentage tolerant population to 3 different doses of INH (0.5, 1 and 5 µg/ml) and RIF (0.1, 0.5 and 1 µg/ml) across 4 different time points (3rd day, 6th day, 9th day and 12th day) in H37Rv infected ADSCs. Experiment was done three times. Error bar represents S.D.

3rd part: Host ABC transporters ABCC1 and ABCG2 play a key role in bacterial drug tolerance

- Microarray data could be presented in a way where authors can highlight and show upregulation and downregulation of different genes. And they can point out the specific ones.

We have replotted the microarray results as suggested. The entire excel for expression value and statistics are shown in Table S1.

- Fig 1J, I am not convinced of the upregulations of ABCC1 and ABCG2 based on those histograms, maybe if they write down the percentage of increase in ABCC1 and ABCG2 receptors? It really does not look to be statistically significant.

We have added the numbers to the histograms reflecting the MFI. The data represent 10,000 cells and while there is a small shift in MFI, the entire population has shifted, suggesting a consistent pattern.

Figure 1(J) Line histogram of surface and intracellular (I.C.) staining of ABCC1/MRP-1 and ABCG2/BCRP in uninfected (Blue line) and in H37Rv-infected ADSCs (Green line), 6 days post-infection. Red line represents the isotype control. Numbers represent MFI of individual colored histogram.

- Line 131, authors have not mentioned the timeline and dose of H37Rv infection, cell harvesting, and staining

Thank you for highlighting. Dose of infection throughout this study used was 1:10, which we have clearly stated in the revised methods section. The timeline for this particular experiment (ABCC1 and ABCG2 surface/Intracellular staining) was 6th Day post-infection, which is indicated in the legends to this figure.

- Supplementary figure S2G and H, Please show the same scale in y-axis

We have plotted them with same scale. These are Fig. S3G and S3H in the revised manuscript.

Figure S3(G-H) CFU assay of *Mtb* burden on 6th day in H37Rv infected ADSCs after 24 hours prior treatment with different doses of MK571 (G) and 25 µg/ml of novobiocin (H).

4th part: Role of lysosomal function in mycobacterial drug tolerance in ADSCs

- Results still do not provide a valid justification of fig 1, why non-drug resistant *Mtb* is increasing after inhibition of ABCC1 and ABCG2. A better explanation of why BafA1, 3MA, and Chloroquine inhibition increase CFUs of non-drug resistant *Mtb*. It is still confusing.

There are two parts to this point. First is the effect of ABCC1 and ABCG2 inhibition on *Mtb* survival. The results indicate ABCC1 and ABCG2 are likely to be directly involved in bacterial killing in ADSCs and therefore upon inhibition helps in an increase in the bacterial CFU. While we could not develop this aspect further and we do not understand the mechanism, presence of ABCC1 and ABCG2 on *Mtb* phagosome supports the possibility of their involvement in bacterial killing.

It is to be noted that the pro-bacterial effect of efflux pump inhibition is seen only in the absence of anti-TB drugs. In the presence of antibiotics, efflux pump inhibition aid antibiotic-mediated bacterial killing leading to reduction in drug-tolerant population.

Secondly, similar to the results observed with efflux pump inhibition, inhibition of lysosomal function directly (by BafA1 or CQ) or indirectly (with 3MA through autophagy) helps better survival of *Mtb* suggesting lysosomal killing mechanism in ADSCs. Here too, the pro-bacterial effects are limited to antibiotic free condition. In the presence of anti-TB drugs, inhibitors of lysosomal function aid antibiotic-mediated bacterial killing.

Together these observations suggest that conditions which otherwise help *Mtb* to evade host anti-bacterial mechanisms, also sensitize the bacteria to anti-TB drugs. There are reports, which suggest dependence of antibiotics on pH for bacterial killing ². We have discussed these points further in the revised manuscript.

In conclusion, we believe, our results are consistent with the inference that lysosomal function helps bacterial killing in ADSCs.

5th part: Effect of inflammatory cytokines IFN γ and TNF α on drug tolerance within ADSCs

- Line 175, Could you please provide more references for this statement?
- Line 180, Is it figure 2D?

While there are no additional references as discretely showing the relationship between immune activation and drug tolerance, several studies indirectly point to the veracity of this statement. For example, macrophage residence itself was shown to impart antibiotic tolerance in *Mtb* ³. Secondly, there is a relationship between pH and antibiotic mediated bacterial killing ². Finally, in a recent study, a single compound was able to inhibit mycobacterial tolerance to oxidative stress, acid stress and drug stress. The compound identified through a screening study, acts on mycobacterial respiration ⁴. Most of these stresses are exacerbated in immune activated cells, thereby supporting the statement above. We have added these references to the discussion section.

- Fig 2E and 2F, why concentration and units of measurement are different? If this is the case, then we cannot compare between these two (Line 184).

In the revised manuscript, we have used same units of measurements to show doses of IFN γ and TNF α .

- Line 193-194, How can we say that *Mtb* is rescued from cytokine-mediated killing?

This statement refers to the effect of IFN γ R1 or anti-TNF α antibody on the killing of *Mtb* in THP-1 macrophages (Fig. 2H) when treated with inflammatory cytokines like IFN γ or TNF α . Neutralizing the cytokine resulted in better survival of *Mtb* within macrophages, thereby rescuing them from cytokine-mediated killing. This supports our inference that *Mtb* is rescued from cytokine mediated killing when cytokines are neutralized.

Figure 2(I) Representative confocal images of PKH67-labeled H37Rv and BCG infected ADSCs on 3rd day post-infection, co-stained with LysoTracker Red and LAMP-1 antibody.

6th part: Analysis of intracellular niches of *Mtb* shows classic phagosome maturation dynamics in ADSCs

- Fig S3A, please mention the scale and magnification.

Thank you for pointing it out. We have included these in the revised manuscript.

- Fig 2K, Line 215-216, It is mentioned that BCG is present mainly in LysoTracker-LAMP1 double positive compartment, but from the figure, it seems like the highest percentage of BCG is in LysoTracker compartment.

In the revised manuscript we provide more representative images for this result.

- Line 223, Fig S2E, For ABCC1, *Mtb* seems to colocalize but I am not convinced with ABCG2.
- Line 226-227, How it is giving an impression that ABCC1 is directly involved in killing?

We performed additional experiments and have revised this figure with a more representative image.

Figure S4(E) PKH67 labeled H37Rv infected ADSCs stained with ABCG2 (orange) along with LysoTracker red. Scale bars, 10 μm .

The role of ABCC1 or ABCG2 in the killing of *Mtb* so far is supported by only one observation that inhibition of these efflux pump results in an increase in bacterial CFU. Moreover, presence of ABCC1 and ABCG2 on *Mtb* phagosome supports the possibility of their involvement in bacterial killing.

7th part: The lipid mediator PGE2 helps MSCs exhibit pro-bacterial attributes

- Figure S5C, actually shows the opposite of what the authors mention in 272. Celecoxib reduces CFUs in THP1 30% while only 10-15% in MSCs

In the revised manuscript, we have edited this section to more accurately represent this point. For S5C, we used Celecoxib at 50, 150 and 250uM. At 50uM, the effect on *Mtb* CFU in THP-1 appears more that that in ADSCs. At higher doses, the effect is more pronounced in ADSCs (60-80%).

Page 12, Line 280: "Celecoxib was also effective in killing *Mtb* within macrophages however unlike in MSCs, it did not show any dose-dependent killing in macrophages at the tested doses (Fig. S6C)."

8th part: MSCs serve as a niche for *Mtb* during in vivo infection allowing drug tolerance in PGE2 dependent manner

- Please show the group with the aerosol challenge (control)

While we always have aerosol challenge control group, they are never shown in the CFU data. We have added the percent population data for MSC and macrophages from the control animals at week 4 and week 12 in the supplementary figure S7B.

Figure S7(B) Plot showing percent macrophage and MSC population in the lungs of uninfected animals (green) with respect to those in the *Mtb* infected ones (purple). The data from the infected animals are same as shown in Fig. 4.

• When INH+ Celecoxib were given in combination, where the doses of these two 50 and 10mg/kg respectively?

Yes, in combination treatment INH was 10mg/kg and celecoxib was 50mg/kg. We have described it in the figure legend in the revised manuscript.

• Fig 4A, line 300-301, it is not clear from the figure that there are differences in CFUs, maybe if authors present the figure in log scale, things could be clear.

Figure 4(A) Total bacterial CFU in the lungs of C57BL/6 mice infected with H37Rv via aerosol route (~10² bacilli/lung) administered with vector, celecoxib (50 mg/kg), INH (10 mg/kg) or combination of celecoxib with INH (50 mg/kg and 10 mg/kg respectively). Treatments started 4 weeks post-infection and were given every day for next 8 weeks.

Fig 4A is already in log scale. However we have replaced the previous figure with a new one where we have data from two more animal experiments added and differences at least for INH and INH+celecoxib groups are more prominent now.

• Plot fig 4A and 4B please show the figures with the scale same

In the revised manuscript, for both 4A and 4B, we have kept the Y-axis scale at 10⁷.

• Fig 4C, I am not convinced with the markers used for characterizing MSCs and macrophages populations. I would suggest using more markers like CD44+, CD90+, CD105+ and CD45- and CD11B- for MSCs

and Ly6C+ or F4/80+, CD11C-, Ly6G- for macrophages. • Line 136, Fig 4F, how did you count cells?

Figure 4(D) Characterization of the gated macrophage and MSC population with additional cell specific surface marker. Upper panel is for macrophage markers namely Ly6G, CD11c, Ly6C and MHC-II. Lower panel is composed of MSC markers i.e. CD44, CD90 and CD105.

As suggested, we have characterized MSCs and macrophage population with the markers as suggested by this reviewer. These characterizations are included in the Fig. 4D of the revised manuscript. The macrophage population however did show CD11c expression, which is also reported earlier by others⁵.

For cell count, actual number of sorted cells was used.

- Fig 4G, please explain why CFU can decrease at

week 8 and then increase again in case of the group treated with celecoxib?

The observed increase in MSC-resident Mtb population in celecoxib treated animals at 12 weeks time following a decline observed at 8 weeks time could simply reflect the dynamics of recruitment of various immune cells and corresponding functional states during the course of tuberculosis pathogenesis. The dynamics of recruitment of different immune cells including MSCs in tuberculosis granulomas and the dissection of their temporal functionality is a challenging question. At this stage, we have insufficient results to infer on this observation.

Materials and methods:

- Reagents: Please provide all the necessary information about the antibodies like the fluorophore attached, clone etc. Also, few of the antibodies have not been written, please add all.

Thank you for this point. We provide the entire list of antibodies along with relevant details in the materials section of the revised manuscript.

- Please explain where did the H37Rv came from or how it was transfected

H37Rv seed stock was obtained from Colorado State University, USA. GFP-H37Rv was prepared by electroporating virulent H37Rv strain with pMN437-GFPm2 vector (Addgene, 32362) and was maintained at 50 ug/ml hygromycin in 7H9-ADC media.

- Please mention the strain of BCG used in this study.

The BCG (Danish strain) was obtained from University of Delhi, South Campus.

- Why authors have used ADC and not OADC?

ADC is recommended for liquid culture whereas OADC for solid culture. All CFU plating were performed on 7h11 agar plates supplemented with OADC whereas all liquid cultures were performed in 7H9 media supplemented with ADC.

- Line 530, Why 4 hours for macrophage and 12 hours for ADSCs?

This was based on our early standardization experiment where ADSCs were infected for 4, 8 and 12 hours. Only at 12 hours consistently >80% infectivity was achieved.

- At what time were plates counted?

All the CFU plates were counted at 21 days post-plating.

- Which version of FlowJo was used?

We used Flow Jo V10.5.3. It is now added in the revised methods section.

- How many reads were acquired in Illumina?

We used Illumina bead array for microarray experiment. This was not a RNA-seq experiment and therefore we do not have read counts. The raw data is available at GEO database "GSE133803".

- Please mention how mice were euthanized.

Mice were euthanized using CO2 inhalation strategy, which has been approved by the IAEC.

- Tissue processing, it seems like authors have processed lung and spleen in the same way, which is probably not the case. Please write the protocol clearly

We have revised the section on tissue processing from animal experiments.

- Please mention the dilution used for anti-rabbit IgG antibody (Line 634).

We have added these details in the revised manuscript.

- 1 Ciuffreda, M. C., Malpasso, G., Musaro, P., Turco, V. & Gneccchi, M. Protocols for in vitro Differentiation of Human Mesenchymal Stem Cells into Osteogenic, Chondrogenic and Adipogenic Lineages. *Methods in molecular biology (Clifton, N.J.)* **1416**, 149-158, doi:10.1007/978-1-4939-3584-0_8 (2016).
- 2 Bartek, I. L. *et al.* Antibiotic Bactericidal Activity Is Countered by Maintaining pH Homeostasis in Mycobacterium smegmatis. *mSphere* **1**, doi:10.1128/mSphere.00176-16 (2016).
- 3 Adams, K. N. *et al.* Drug tolerance in replicating mycobacteria mediated by a macrophage-induced efflux mechanism. *Cell* **145**, 39-53, doi:10.1016/j.cell.2011.02.022 (2011).

- 4 Flentie, K. *et al.* Chemical disarming of isoniazid resistance in *Mycobacterium tuberculosis*. *Proceedings of the National Academy of Sciences of the United States of America* **116**, 10510-10517, doi:10.1073/pnas.1818009116 (2019).
- 5 Huang, L., Nazarova, E. V., Tan, S., Liu, Y. & Russell, D. G. Growth of *Mycobacterium tuberculosis* in vivo segregates with host macrophage metabolism and ontogeny. *The Journal of experimental medicine* **215**, 1135-1152, doi:10.1084/jem.20172020 (2018).

Reviewers' Comments:

Reviewer #1:

Remarks to the Author:

Major comment: The majority of comments have been well addressed by the authors. The additional work and explanations are appreciated.

Remaining comments: The immunohistochemistry dual stain (red and brown) is much improved but not quite convincing on the computer screen. There are 2 reasons: First - the magnification may be too low, and the authors could consider increasing the magnification to 400x dry objective, or 1000x with oil immersion? Second - negative controls (tissue sections incubated with isotype-matched antibody) seem to be lacking. Negative controls would help determine the non-specific staining.

Reviewer #2:

Remarks to the Author:

My original comment: " Fig4 I: In contrast to Fig4H, Fig4I is not convincing. First larger orange arrows lie on top of smaller arrows, if you zoom into the picture. This is misleading and should be avoided. Irrespective of the arrows the CD73 positive cells cannot be identified with the blue counterstaining within the pictures of the tissue sections provided. This needs to be readdressed in detail."

Author reply: "We have replaced figures 4H and 4I with a much higher resolution images showing clear distinction between the dual staining. The arrows are shown prominently and we believe the histopathology data in the revised manuscript matches up to the desired standard."

- The authors have replaced figures 4H and 4I with novel IHC images with higher resolution. The arrows are now clearly visible. However the histopathology data do not meet the expected standards (s. below).

My original comment: "The authors do state in their text that Ag85 and CD73 are "in close vicinity" but this is in fact several cell diameters away. It appears that in contrast to their BL6 mouse data, the authors do not see CD73 cells harboring Mtb bacteria in human TB lesions. The apparent key question is: Do MSC in the human TB patient harbor Mtb bacteria as the mouse data suggest, or does that differ. In order to address this the authors may consider e.g. a version of a proximity ligation assays (e.g. CD73 vs Ag85) to demonstrate the presence of Mtb within MSC or may use an independent staining protocol, where the signals can be undoubtedly visualized and clearly separated from each other. Another way to address this may be the use of a Ziehl-Neelsen stain followed by IHC for CD73."

Author reply: Previously, the dual staining in human granuloma samples were not as well standardized and therefore we could not really see their co-localization. In the revised manuscript we indeed show clear co-localization between CD73 and Mtb. In addition, we also observe similar co-localization between CD105 (another MSC marker) and Mtb. Therefore we have firmed up our inference here and instead of saying "in close vicinity", we do mention "MSC harboring Mtb could be seen in and around the granulomas".

- The overall quality with regard to the provided IHC data has improved. However, the authors did not appropriately address the question whether MSC in the human TB patient harbor Mtb bacteria as the mouse data suggest. The presented stainings are not convincing. I disagree with the authors that the provided data allow that the statement that "MSC harboring Mtb could be seen in and around the granulomas".

- o The use of a red (CD73 and CD105) and brown (MtbAg85B) detection system within one tissue section does not allow the the identification of double positive cells in the provided data sets. The independent signals cannot be undoubtedly visualized and therefore cannot be clearly separated from each other.
- o In contrast to the very clear and prominent detection of CD73 and CD105 by IHC, the Ag85B stainings are of poor quality. The brown staining for Ag85B is very weak. Prominently stained Ag85B-positive cells cannot be detected in the provided figures, thus double positive cells cannot be identified. The black arrows do not clearly identify Ag85B-positive cells.
- o The authors should either provide additional immunofluorescence (IF) double stainings for CD73/Ag85B and CD105/Ag85B or perform a Ziehl-Neelsen stain followed by IHC for CD73 or CD105 as suggested previously.

Reviewers' comments:

Reviewer #1 (Remarks to the Author):

Major comment: The majority of comments have been well addressed by the authors. The additional work and explanations are appreciated.

Thank you for appreciating our efforts on revising the manuscript.

Remaining comments: The immunohistochemistry dual stain (red and brown) is much improved but not quite convincing on the computer screen. There are 2 reasons: First - the magnification may be too low, and the authors could consider increasing the magnification to 400x dry objective, or 1000x with oil immersion? Second - negative controls (tissue sections incubated with isotype-matched antibody) seem to be lacking. Negative controls would help determine the non-specific staining.

To address the above concerns with the IHC, we further standardized the dual staining and the revised manuscript has now newer sets of IHC images, which are much more convincing and easy to interpret. More specifically:

a) We realized that red and brown chromogens for dual IHC were not sufficiently distinct.

Figure 5 (C) On high power examination of two independent human lung biopsies, Ag85B positive organisms (brown color) are seen inside the histiocytes (black arrows), CD73⁺ cells are stained with blue chromogen (green arrows) and the cells showing both positivity for CD73⁺ and Ag85B⁺ organisms have been represented by red arrows (x 400). Insets below show the corresponding magnified CD73⁺ cells showing positivity of Ab85B staining.

b) We have used isotype controls in the revised manuscript and the images are shown in figure S7C. Also, we describe below our efforts to perform IF staining to address the other reviewer's comments. We performed isotype controls for IF staining as well. We thank the reviewer for this excellent suggestion.

Figure S7 (C) For antibody controls, skin biopsy from a patient with known *Lupus vulgaris* infection was included. The left panel show isotype controls with no brown or blue positivity. Fast red stain was used to highlight the background stromal cells. (x 100).

c) In addition, we also performed IFA on these sections using both CD73 and CD105 for co-staining with Ag85B. The isotype controls and positive controls were

Figure S7 D Panels from left to right shows isotype control used, positive control used for two colored immunofluorescence staining using Ag85B (green) with CD73 stains (red), and Ag85B (green) with CD105 (red) stains (Scale bar: 10 μ m).

done on the skin TB biopsy samples (Fig. S7D). In human lung TB granuloma sections, at 100x oil immersion objective, we find specific cells showing Ag85b positivity and CD73 or CD105 positivity (Fig. 5D and 5E).

Figure 5(D) Immunofluorescence staining performed on formalin fixed paraffin embedded (FFPE) tissue of human lung biopsies from patients with known tuberculosis show green fluorescence for Ag85B⁺ (green arrows), red fluorescence for CD73⁺ cells (red arrows) and colocalization signals are marked with white arrows. The strong co-localization area is shown in the yellow inset and magnified in the panel at the right. In the further right panel, corresponding green and red channel fluorescence is shown. (E) FFPE tissue processed for dual IF staining show Ag85B⁺ only (green arrows), CD105⁺ MSCs (red arrow) and cells positive for both CD105 and Ag85B (white arrow). The strong co-localization area is shown in the yellow inset and magnified in the panel at the right. In the further right panel, corresponding green and red channel fluorescence is shown. Scale bar is 10 μ m.

Taken together, through these new IHC and IF assays we are now convinced that human TB granulomas indeed have MSCs harboring the *Mtb*.

Reviewer #2 (Remarks to the Author):

My original comment: ” Fig4 I: In contrast to Fig4H, Fig4I is not convincing. First larger orange arrows lie on top of smaller arrows, if you zoom into the picture. This is misleading and should be avoided. Irrespective of the arrows the CD73 positive cells cannot be identified with the blue counterstaining within the pictures of the tissue sections provided. This needs to be readdressed in detail.”

Author reply: “We have replaced figures 4H and 4I with a much higher resolution images showing clear distinction between the dual staining. The arrows are shown prominently and we believe the histopathology data in the revised manuscript matches up to the desired standard.”

- **The authors have replaced figures 4H and 4I with novel IHC images with higher resolution. The arrows are now clearly visible. However the histopathology data do not meet the expected standards (s. below).**

Thank you for acknowledging the effort in improving the IHC images.

My original comment: “The authors do state in their text that Ag85 and CD73 are “in close vicinity” but this is in fact several cell diameters away. It appears that in contrast to their Bl6 mouse data, the authors do not see CD73 cells harboring Mtb bacteria in human TB lesions. The apparent key question is: Do MSC in the human TB patient harbor Mtb bacteria as the mouse data suggest, or does that differ. In order to address this the authors may consider e.g. a version of a proximity ligation assays (e.g. CD73 vs Ag85) to demonstrate the presence of Mtb within MSC or may use an independent staining protocol, where the signals can be undoubtedly visualized and clearly separated from each other. Another way to address this may be the use of a Ziehl-Neelsen stain followed by IHC for CD73.”

Author reply: Previously, the dual staining in human granuloma samples were not as well standardized and therefore we could not really see their co-localization. In the revised manuscript we indeed show clear co-localization between CD73 and Mtb. In addition, we also observe similar co-localization between CD105 (another MSC marker) and Mtb. Therefore we have firmed up our inference here and instead of saying “in close vicinity”, we do mention “MSC harboring Mtb could be seen in and around the granulomas”.

- **The overall quality with regard to the provided IHC data has improved. However, the authors did not appropriately address the question whether MSC in the human TB patient harbor Mtb bacteria as the mouse data suggest. The presented stainings are not convincing. I disagree with the authors that the provided data allow that the statement that “MSC harboring Mtb could be seen in and around the granulomas”.**

We appreciate the critical observations made by the reviewer, which led us to further step up our co-staining protocols. With the new IHC images and through the IF

staining, we confidently state that human TB granulomas indeed show presence of MSCs that harbor the bacteria. The specific responses to the concerns are as follows:

o The use of a red (CD73 and CD105) and brown (MtbAg85B) detection system within one tissue section does not allow the the identification of double positive cells in the provided data sets. The independent signals cannot be undoubtedly visualized and therefore cannot be clearly separated from each other.

Figure 5 (C) On high power examination of two independent human lung biopsies, Ag85B positive organisms (brown color) are seen inside the histiocytes (black arrows), CD73⁺ cells are stained with blue chromogen (green arrows) and the cells showing both positivity for CD73⁺ and Ag85B⁺ organisms have been represented by red arrows (x 400). Insets below show the corresponding magnified CD73⁺ cells showing positivity of Ab85B staining.

Thank you for your concerns on using red and brown detection system in the previous version. In the revised manuscript, we have used blue and brown detection system, which now can be visualized individually with ease and therefore makes it easy to notice the double positive cells. We have also zoomed-up specific regions from the images to further highlight the dual positivity (Fig. 5C).

o In contrast to the very clear and prominent detection of CD73 and CD105 by IHC, the Ag85B stainings are of poor quality. The brown staining for Ag85B is very weak. Prominently stained Ag85B-positive cells cannot be detected in the provided figures, thus double positive cells cannot be identified. The black arrows do not clearly identify Ag85B-positive cells.

The brown staining for Mtb Ag85B is very distinct in the revised images, as can be seen in the figure 5C above.

o The authors should either provide additional immunofluorescence (IF) double stainings for CD73/Ag85B and CD105/Ag85B or perform a Ziehl-Neelsen stain followed by IHC for CD73 or CD105 as suggested previously.

As suggested by the reviewer, we tried both ZN-stain combined with CD73/CD105 IHC and IF staining separately. While we were not as successful in standardizing the ZN stain in combination with IHC, we got excellent slides for IF, which we visualized under confocal microscope using 100x objective (oil immersion). The IFA images (shown in Fig. 5D and 5E) unequivocally show presence of *Mtb* within MSCs in human lung granulomas.

Figure 5 (D) Immunofluorescence staining performed on formalin fixed paraffin embedded (FFPE) tissue of human lung biopsies from patients with known tuberculosis show green fluorescence for Ag85B⁺ (green arrows), red fluorescence for CD73⁺ cells (red arrows) and colocalization signals are marked with white arrows. The strong co-localization area is shown in the yellow inset and magnified in the panel at the right. In the further right panel, corresponding green and red channel fluorescence is shown. (E) FFPE tissue processed for dual IF staining show Ag85B⁺ only (green arrows), CD105⁺ MSCs (red arrow) and cells positive for both CD105 and Ag85B (white arrow). The strong co-localization area is shown in the yellow inset and magnified in the panel at the right. In the further right panel, corresponding green and red channel fluorescence is shown. Scale bar is 10 μm.

Taken together, through these new IHC and IF assays we are now convinced that human TB granulomas indeed have MSCs harboring the *Mtb*.

Reviewers' Comments:

Reviewer #2:

Remarks to the Author:

With regard to the revised manuscript provided by the authors I like to state that the authors have thoroughly and now appropriately addressed my latest comments. In particular based the provided IF data, I do now accept the statement that Ag85 is present in some CD73 and CD105 positive cells. However the authors should refrain from exaggerating and overinterpreting their findings. They should "disarm" and better refine the wording:

- Terms as "Routinely" and "frequently" ask for quantification, which I do not ask for. Thus, please omit these adverbs and let the experienced reader decide.

- The statement "can revolutionize" reflects the excitement and the enthusiasm of the authors but I do not consider this to be appropriate with regard to the science. This is really to much.

Thus I suggest to end the abstract as follows: "Moreover, MSCs are observed in and around human tuberculosis granulomas, harboring Mtb bacteria. We, therefore, propose, targeting the unique immune-privileged niche, provided by MSCs to Mtb, can have a major impact on tuberculosis prevention and cure."

Norbert Reiling

Response to the reviewer's comments:

Reviewer #2 (Remarks to the Author):

With regard to the revised manuscript provided by the authors I like to state that the authors have thoroughly and now appropriately addressed my latest comments. In particular based the provided IF data, I do now accept the statement that Ag85 is present in some CD73 and CD105 positive cells. However the authors should refrain from exaggerating and overinterpreting their findings. They should "disarm" and better refine the wording:

- Terms as "Routinely" and "frequently" ask for quantification, which I do not ask for. Thus, please omit these adverbs and let the experienced reader decide.
- The statement "can revolutionize" reflects the excitement and the enthusiasm of the authors but I do not consider this to be appropriate with regard to the science. This is really to much.

Thus I suggest to end the abstract as follows: "Moreover, MSCs are observed in and around human tuberculosis granulomas, harboring Mtb bacteria. We, therefore, propose, targeting the unique immune-privileged niche, provided by MSCs to Mtb, can have a major impact on tuberculosis prevention and cure."

We are pleased to note that the reviewer is satisfied with our efforts to address the very important concerns raised earlier. We agree with the reviewer's suggestions on the abstract and we have accordingly revised the abstract.